

# Atmospheric processing and aerosol aging responsible for observed increase in absorptivity of long-range transported smoke over the southeast Atlantic

Abdulamid A. Fakoya[1], Jens Redemann[1], Pablo E. Saide[2,3], Lan Gao[1], Logan T. Mitchell[1], Calvin Howes[3], Amie Dobracki[4], Ian Chang[5], Gonzalo A. Ferrada[6,7], Kristina Pistone[8,9], Sam E. Leblanc[8,9], Michal Segal-Rozenhaimer[8,9,13], Arthur J. Sedlacek III[4], Thomas Eck[10,11], Brent Holben[11], Pawan Gupta[11], Elena Lind[11], Paquita Zuidema[12], Gregory Carmichael[14,15] and Connor J. Flynn[1]

[1]School of Meteorology, University of Oklahoma, Norman, Oklahoma, USA

[2]Institute of the Environment and Sustainability, University of California, Los Angeles, Los Angeles, California, USA

[3]Department of Atmospheric and Oceanic Sciences, University of California, Los Angeles, Los Angeles, California, USA

[4]Environmental and Climate Sciences Department, Brookhaven National Laboratory, Upton, New York, USA

[5]Department of Earth, Environmental, and Geographical Sciences, The University of North Carolina at Charlotte, Charlotte, North Carolina, USA

[6]Cooperative Institute for Research in Environmental Sciences, University of Colorado, Boulder, Colorado, USA

[7]Global Systems Laboratory, National Oceanic and Atmospheric Administration, Boulder, Colorado, USA

[8]Bay Area Environmental Research Institute, Moffett Field, California, USA

[9]NASA Ames Research Center, Moffett Field, California, USA

[10]University of Maryland Baltimore County, Baltimore, Maryland, USA

[11]NASA Goddard Space Flight Center, Greenbelt, Maryland, USA

[12]Department of Atmospheric Sciences, Rosenstiel School, University of Miami, Miami, Florida, USA

[13]Department of Geophysics, Porter School of the Environment and Earth Sciences, Tel-Aviv University, Tel-Aviv, Israel

[14]Center for Global and Regional Environmental Research (CGRER), University of Iowa, Iowa City, Iowa, USA

[15]Department of Chemical and Biochemical Engineering, University of Iowa, Iowa City, Iowa, USA

*Correspondence to*: Abdulamid A. Fakoya (abdulamid.fakoya@ou.edu) and Jens Redemann (jredemann@ou.edu)





**Abstract.**

Biomass burning aerosols (BBA) from agricultural fires in southern Africa contribute about one-third of global carbonaceous aerosol load. These particles have strong radiative effects in the southeast Atlantic (SEA), which depend on the radiative contrast between the aerosol layer in the free troposphere (FT) and the underlying cloud layer. However, there is large disagreement in model estimates of aerosol-driven climate forcing due to uncertainties in the vertical distribution, optical properties, and lifecycle of these particles.

This study applies a novel method combining remote sensing observations with regional model outputs to investigate the aging of the BBA and its impact on the optical properties during transatlantic transport from emission sources in Africa to the SEA. Results show distinct variations in Ångstrom exponent (AE) and single scattering albedo (SSA) as aerosol age. Near the source, fresh aerosols are characterized by low mean SSA (0.84) and high AE (1.85), indicating smaller, highly absorbing particles. By isolating marine contributions from the total column during BBA transport across the SEA, our analysis reveals an initial decrease in BBA absorptivity, with mean FT SSA of 0.87 after 6–7 days, followed by increased absorptivity with mean FT SSA of 0.84 after 10 days, suggesting enhanced absorption due to chemical aging.

These findings indicate that BBA becomes more absorbing during extended transport across the SEA, with implications for reducing model uncertainties. Our remote sensing-based results agree well with previous in situ studies and offer new insights into aerosol-radiation interactions and the energy balance over the SEA.





## 1 Introduction

Atmospheric aerosols play a crucial role in Earth's energy balance and climate system. Aerosols perturb the vertical temperature structure, altering the atmospheric stability and overall radiation balance through interactions with sunlight via scattering and absorption, known as aerosol-radiation interactions (ARI) (Ackerman et al., 2000; Boucher, 2015; Santer et al., 1996; Hansen et al., 1997; Koch and Del Genio, 2010; Samset et al., 2018). Aerosols also serve as cloud condensation nuclei (CCN) or ice-nucleating particles (INPs) that influence cloud microphysics and modify cloud reflectivity (Twomey, 1974, 1977) and lifespan (Albrecht, 1989) through aerosol-cloud interactions (ACI) (Ackerman et al., 2000; Boucher, 2015). The resulting impacts of ARI and ACI on climate are cumulatively quantified as climate forcing, and its magnitude and sign depend on various factors, including particle size, composition, concentration, mixing state, optical properties, and vertical distribution at a given location.

Single-scattering albedo (SSA), the ratio of aerosol scattering to aerosol extinction, an indicator of aerosol absorptivity, is an important parameter in deriving the radiative effects of aerosols (Chylek and Wong, 1995; Takemura et al., 2002; Bergstrom et al., 2007; Satheesh et al., 2010). Non-absorbing aerosols have SSA of 1, whereas absorbing aerosols have lower values (Moosmüller et al., 2012). The magnitude of the direct aerosol radiative effect depends on the interplay between the radiative properties of aerosols and the underlying scene (Keil and Haywood, 2003; Chand et al., 2009). Non-absorbing aerosols more effectively elevate the scene albedo in regions with inherently darker scenes, whereas absorbing aerosols decrease the albedo most strongly over brighter surfaces, with a comparatively lesser impact over darker surfaces (Mishra et al., 2015; Bellouin et al., 2020). On the global average for radiative forcing, absorbing aerosols exert a cooling effect at the top of the atmosphere. In clear-sky maritime scenarios, this radiative forcing is negative. However, when clouds are present in certain regions such as the southeast Atlantic (SEA), the interaction between aerosol and cloud layers can lead to positive radiative forcing, contingent upon the vertical distribution of the aerosols relative to the clouds (Haywood and Shine, 1997; Keil and Haywood, 2003; Chand et al., 2009).

Biomass burning (BB) is the largest source of carbonaceous aerosols globally (Bowman et al., 2009; Vermote et al., 2009; Bond et al., 2013), emitting significant amounts of brown carbon (BrC), light absorbing organic aerosols, and up to 40% black carbon (BC), the strongest light-absorbing aerosol component, into the atmosphere (Bond et al., 2004; Andreae and Gelencsér, 2006; Hopkins et al., 2007; Boucher et al., 2013). The composition of BB aerosols (BBA) is highly variable and often depends on the fuel type and burning conditions, with the less efficient smoldering phase of fires having more organic aerosols (OA) and less BC than the flaming phase (Levin et al., 2010; Liu et al., 2014; Laskin et al., 2015; Jen et al., 2019; Zhou et al., 2017; Collier et al., 2016). BrC exhibits lower mass absorption efficiency compared to BC (Alexander et al., 2008; Jimenez et al., 2009). However, in fires where BrC and BC are co-emitted, the absorption properties of BBA can be influenced by the complex composition which consists of diverse organic compounds capable of oxidation and photochemistry. These reactions, along with particle morphology and the mixing state of the carbonaceous constituents as well as the condensation of gases and other coating materials upon the BC cores as in (Lack and Cappa, 2010; Dobracki et al., 2023) can modify the absorptive characteristics and atmospheric residence time of the aerosols (Feng et al., 2013; Saleh et al., 2015; Feng et al., 2021). Given the projected



global increase in BB events (Keywood et al., 2013; Jones et al., 2022), an understanding of BBA and their spatial and temporal evolution becomes essential to improve the estimate of their climate forcing.

Southern Africa contributes approximately 35% of the global biomass burning emissions (Van Der Werf et al., 2010; Granier et al., 2011; Redemann et al., 2021). Annually, between June and October, these emissions are

transported westward over the SEA (Holanda et al., 2020) where they overlie a semi-permanent deck of stratocumulus (Sc) cloud (Figure 1) and occasionally mix into the marine boundary layer (MBL) (Zhang and Zuidema, 2021). Consequently, the region is characterized by heavy periodic loadings of BBA, which represent the global maximum of aerosol optical depth (AOD) above clouds (Waquet et al., 2013; Adebiyi et al., 2015).

BBA in this region typically have SSA values ranging between 0.7 and 0.9 from observations (Dubovik et

al., 2002; Haywood et al., 2003; Eck et al., 2013). These values generally increase from July to November (Eck et al., 2013), with an average value of 0.85 during the burning season (Leahy et al., 2007; Eck et al., 2013; Pistone et al., 2019; Dobracki et al., 2023), indicating their significant ability to absorb sunlight. BBA also serve as CCN and account for approximately 65% of the total CCN in the Sc cloud deck of the SEA (Andreae and Rosenfeld, 2008; Dedrick et al., 2024; Che et al., 2021; Lenhardt et al., 2023). The warming effect and the ACI by BBA are not well represented

in Earth System Models (ESMs), and the SEA region exhibits a large model-to-model divergence of climate forcing due to aerosols (Sakaeda et al., 2011; Stier et al., 2013; Mallet et al., 2020; Che et al., 2021; Haywood et al., 2021). Reliable quantification of the climatic effects of BBA requires an accurate representation of aerosol properties and their vertical and horizontal distributions in models.

Despite recent studies that have documented the transport and characterized the chemical, optical and

microphysical properties of the southern African BBA transport (Denjean et al., 2020a; Pistone et al., 2019; Wu et al., 2020; Baars et al., 2021; Holanda et al., 2020; Denjean et al., 2020b; Vakkari et al., 2018; Dobracki et al., 2023), it is still not well understood how the properties of these aerosols evolve during long-range transport. The southern African BBA have been associated with the BC-rich pollution layer above the Amazonian basin following extended transport over the SEA (Holanda et al., 2020). Variations in the chemical composition of BBA at different altitude above the

Atlantic have been shown to influence the optical properties of aged BBA (Wu et al., 2020), with low SSA values attributed to the presence of strongly absorbing refractory black carbon (rBC), and minimal contribution from BrC (Denjean et al., 2020b). The mixing state of rBC particles (Denjean et al., 2020b; Sedlacek et al., 2022) as well as the accumulation of a non-absorbing shell by rBC (Redemann et al., 2001) strongly affect the SSA of BBA. Knowledge of the spatial and temporal evolution of BBA, especially during long-range transport is key to understanding the

atmospheric processes affecting their lifecycle and their contribution to climate forcing.

While the aging of BBA has recently been parameterized in global climate models (GCMs) (Konovalov et al., 2021; Nascimento et al., 2021), accurately representing the optical and microphysical characteristics of these particles in models remains a challenge due to the intricate chemical and physical processes involved (Brown et al., 2021) and because there have been limited studies on aged BBA. Most research on the evolution and aging of BBA

has focused on BBA that has been sampled near the source or in short-term laboratory experiments (Reid et al., 1998; Bond et al., 2006; Kleinman et al., 2020; Liu et al., 2021; Feng et al., 2021; Sedlacek et al., 2022). Due to the lack of observations on longer time scales, there exists a gap in our understanding of how the optical properties of BBA



change during extended transport, such as that over the SEA. Investigating the evolution of aged BBA is, therefore, crucial for improving model capabilities for representing their optical and radiative properties.

The overarching goal of this research is to examine the evolution of BBA emitted during the annual burning season in southern Africa using remote-sensing observations during their transport across the SEA. Studies on BBA in the region have focused primarily on either continental southern Africa (Abel et al., 2003; Eck et al., 2003; Queface et al., 2011; Eck et al., 2013) or over the ocean (Meyer et al., 2013; De Graaf et al., 2014; Zuidema et al., 2018; Pistone et al., 2019; Redemann et al., 2021). Measurements over the ocean from the ORACLES (ObseRvations of Aerosols
above CLouds and their intEractionS) campaign dataset highlighted the connection between BBA composition and absorption, showing a decrease in SSA concurrent with the loss of OA coating on rBC particles, as shown by Sedlacek et al. (2022) and Dobracki et al. (2023). However, our study seeks to expand upon these findings by analyzing aerosol absorption along the entire transport pathway of BBA from the emission source to SEA region (Figure 1). Here, we develop a new methodology that integrates remotely sensed observations from AERONET (AErosol RObotic
NETwork) in the BB emission region and 4STAR (Spectrometers for Sky-Scanning, Sun-Tracking Atmospheric Research) over the ocean, collected during the ORACLES campaign. These observations are combined with aerosol age estimates outputs from the WRF-AAM (Weather Research and Forecasting coupled with Aerosol Aware Microphysics module) regional model (Saide et al., 2016) and aerosol properties derived from the WRF-CAM5 (Weather Research and Forecasting coupled with the Community Atmosphere Model version 5). WRF-CAM5 has
previously been used to document the chemical composition, hygroscopicity and aerosol-cloud interactions of BBA in the ORACLES campaign region (Howes et al., 2023). This approach enables a comprehensive assessment of the aging of the smoke plume and its impact on optical properties. Particularly, we use the observation of BBA from ORACLES campaign during the burning season of September 2016, August 2017 and October 2018, to investigate the changes in SSA of BBA during their transatlantic transport, spanning the phases of transport over land and, subsequently, over
the SEA.  Furthermore, using remote sensing to retrieve SSA offers the advantage of sampling aerosols in their ambient state and can be replicated to other domains where AERONET observations are available.



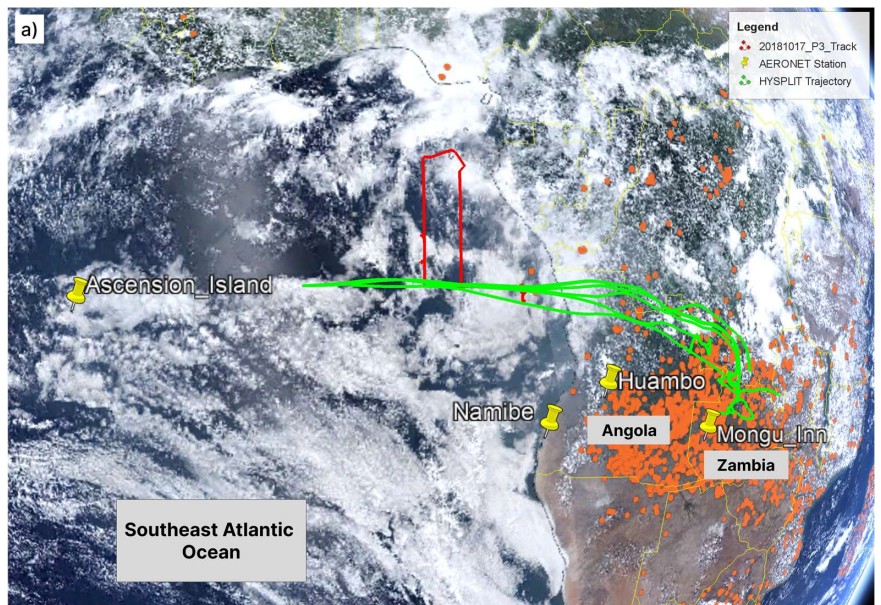

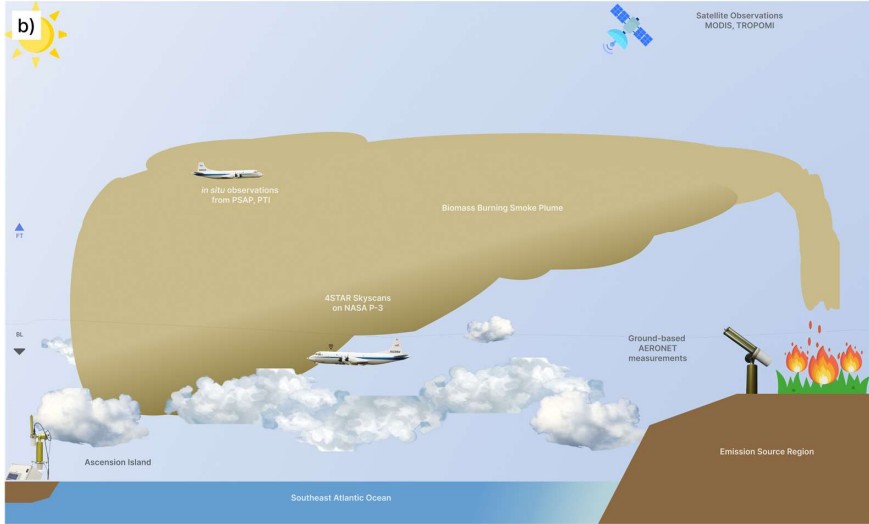

**Figure 1:**
**(a): Satellite image showing the southeast Atlantic Ocean covered by the stratocumulus cloud deck with smoke being advected over it. The smoke is being transported, evident by the 7-day backward HYSPLIT trajectory (green) ending on October 18, 2018, from numerous fires (orange dots) in continental southern Africa. Yellow icons represent AERONET stations selected for this study while the red line represents the P-3 flight tracks on October 17, 2018, showing the intersection with the smoke transport trajectory.**
**(https://worldview.earthdata.nasa.gov/, https://www.ready.noaa.gov/HYSPLIT_traj.php, https://aeronet.gsfc.nasa.gov/, https://earth.google.com).**
**(b): Schematic of the collocation of observations in the southeast Atlantic region during the ORACLES mission. Satellite observation from Moderate Resolution Imaging Spectroradiometer (MODIS) and TROPOspheric Monitoring Instrument (TROPOMI). Airborne observations from 4STAR, ground-based observations from AERONET, in situ measurement from**



**Particle Soot Absorption Photometer (PSAP) and Photo-Thermal Interferometric (PTI) Particle Absorption Monitor. Measurements from 4STAR and AERONET are presented in this study.**

**2 Data and methods**

**2.1 NASA ORACLES campaign**

A comprehensive overview of the ORACLES campaign is presented by Redemann et al. (2021); Here, we summarize the methods and data analysis techniques that are relevant to this study. The campaign provided process-level understanding of aerosol effects in the SEA that can be applied in the parameterization of ACI and ARI in ESMs (Redemann et al., 2021). The campaign overlapped with other field experiments (Zuidema et al., 2016) – CLARIFY-

2017: Cloud-Aerosol-Radiation Interactions and Forcing for Year 2017 (Haywood et al., 2021), LASIC: Layered Atlantic Smoke Interactions with Clouds (Zuidema et al., 2018), AEROCLO-sA: AErosols, RadiatiOn and CLOuds in southern Africa (Formenti et al., 2019) – in a synergistic effort to determine the influence of southern African BBA on cloud properties and the energy balance in the SEA region. The ORACLES campaign occurred between 2016 and 2018 with field deployments in Walvis Bay, Namibia (September 2016), and in São Tomé and Principe (August 2017,

and October 2018). The NASA P3-B aircraft was home to a suite of in-situ and remote-sensing instruments including the 4STAR (Dunagan et al., 2013) and the ER-2 (a high-altitude aircraft) was home to a suite of additional remote sensing instruments. A total of 56 research flights:12 from the ER-2, only in 2016, and 44 from the P-3B across the three deployments, shown in Figure 2, with over 450 science flight hours collected data on aerosol optical properties (Pistone et al., 2019; Redemann et al., 2021). This study uses 4STAR retrievals of BBA properties (Table 1) from all

three campaign deployments made on the P-3B aircraft, to study the evolution of light absorption properties of BBA.

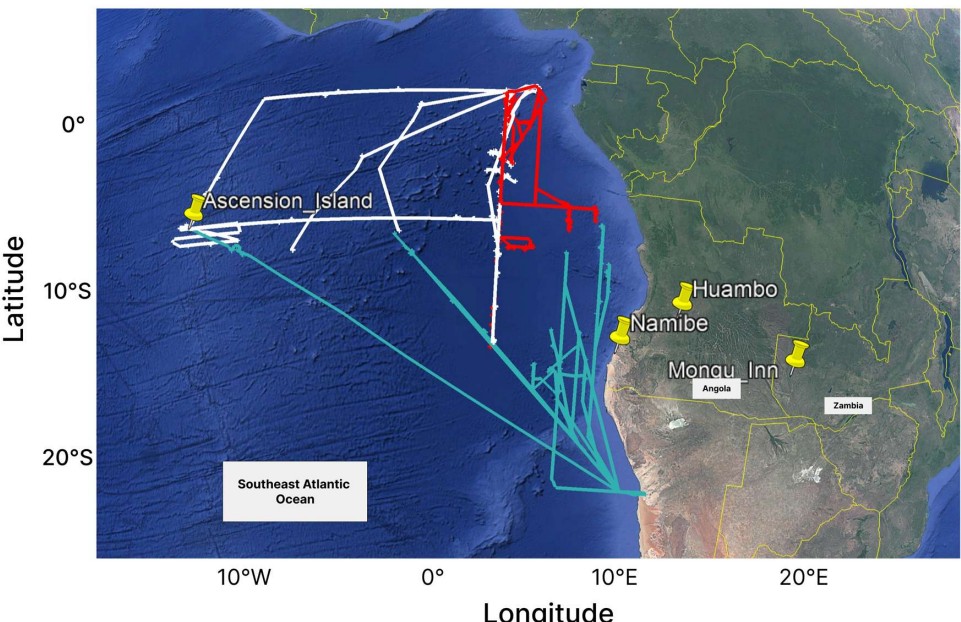

**Figure 2: Map of the SEA showing the NASA P-3B flight tracks during ORACLES 2016 (cyan), 2017 (white) and 2018 (red) for observation days analyzed in this study. The ORACLES aircraft deployed out of Walvis Bay, Namibia in 2016 and São Tomé and Príncipe in 2017 and 2018. Regional AERONET stations are identified using yellow icons. Background imagery from © Google Earth (https://earth.google.com) - Data SIO, NOAA, U.S. Navy, NGA, GEBCO, Image Landsat / Copernicus.**

### 2.2 Ground-based measurements

AERONET, a global sun photometer network, provides long-term, continuous measurement of AOD and retrievals of aerosol microphysical, optical, and radiative properties for aerosol characterization and validation of satellite retrievals (Holben et al., 1998). In this study, we used AERONET Version 3 inversion products (Table 1) from both almucantar (ALM) and hybrid scans including cloud-screened (level 1.5) and quality assured (level 2.0) data (Giles et al., 2019; Sinyuk et al., 2020) from four AERONET stations: three situated in continental southern Africa, **Mongu Inn (Zambia), Huambo (Angola), Namibe (Angola)**; and one in the maritime SEA - **Ascension Island (United Kingdom Territory)** (Table 2). These stations were carefully selected based on their geographic locations aligning with the trajectory of BBA from continental Africa and over the ocean, as shown in Figure 1. Specifically, column-integrated measurements of AOD, SSA, and extinction Ångstrom exponent (hereafter AE) for September 2016, August 2017, and October 2018 were used, coinciding with the observation periods of ORACLES 2016, 2017, and 2018. The Namibe site was inactive during the ORACLES 2018 campaign, providing data only for 2016 and 2017, while the Huambo site, established in 2017, contributed observations for both 2017 and 2018. At Ascension Island and Namibe, fewer observations were available due to the increased cloud cover along the coast and over the SEA. Therefore, to ensure an adequate sample size at Ascension Island and Namibe, we included level 1.5 data in our



analysis when level 2.0 data was unavailable. More information about the AERONET stations utilized in this study is provided in Table 2.

**Table 1: Summary of observation data products and aerosol properties used in this study.**

| Data | Product | Properties | Data Access |
|------|---------|-----------|-------------|
| **AERONET** | Inversion (level 1.5, 2.0) product version 3: All Observation | Aerosol Optical Depth (AOD) & Single Scattering Albedo (SSA) *440, 675, 870 nm* Extinction Ångstrom Exponent (AE) *440-870 nm* | https://aeronet.gsfc.nasa.gov/ |
| **4STAR** | 4STAR-aeroinv_P3 | Aerosol Optical Depth (AOD) & Single Scattering Albedo (SSA) *500, 675, 870, 975 nm* Extinction Ångstrom Exponent (AE) *500-975 nm* | https://espoarchive.nasa.gov/ archive/browse/oracles/P3/4S TAR-aeroinv |

### 2.3 Airborne measurements

4STAR was integrated on the NASA P-3B aircraft to measure direct solar beam and sky radiances (Dunagan et al., 2013), collecting data in three modes: a sun-tracking mode (Leblanc et al., 2020; Segal-Rosenheimer et al., 2014); a sky-scanning mode (Pistone et al., 2019); and a zenith mode (Leblanc et al., 2015). 4STAR is unique in its ability to perform AERONET-like measurements from an aircraft, making it suitable for studying remote regions that AERONET does not cover. It also frequently offers a more comprehensive observational dataset through its deployment alongside in-situ instrumentation, as in the ORACLES campaign. This study primarily examines observations made in the sky-scanning mode, in which, analogous to AERONET, the 4STAR instrument performs ALM scanning to measure the angular distribution of brightness in the sky. In addition, retrievals from principal plane (PPL) scans are also used, as ORACLES flights largely occurred near solar noon, limiting the angular range of ALM scans. As such, we selectively considered both ALM and PPL scans if they met specific quality control (QC) criteria. Here, we processed 4STAR sky scans using the QC criteria from Mitchell et al. (2023). These criteria were adapted from Pistone et al. (2019) for four-wavelength 4STAR retrieval of ORACLES 2016-2018 and serve as a proxy for AERONET level 1.5 aerosol inversion standards. The criteria are: (1) AOD (400 nm) > 0.2, (2) altitude difference < 50 m, (3) sky error < 10%, (4) minimum scattering angle < 6°, (5) maximum scattering angle > 50°, (6) mean scattering angle difference < 3° (between 3.5 and 30°), (7) maximum scattering angle difference < 10° (between 3.5 and 30°), (8) roll standard deviation < 3°, (9) passes retrieval boundary test - ensuring that the retrieval is within limits of parameter space, and (10) maximum altitude < 3000 m. A summary of the valid QC'd 4STAR retrievals of SSA, AOD, AE (Mitchell et al., 2023) used from the ORACLES dataset (Oracles, 2020) for all three deployments is given in Table 1.



**Table 2: Site information and total observation count for AERONET and 4STAR.**

| Site Name | Latitude | Longitude | Elevation (m) | Number of Observations *(2016, 2017, 2018)* |
|---|---|---|---|---|
| Ascension Island | -7.97 | -14.41 | 30 | *77 (17, 53, 7)* |
| Namibe | -15.15 | 12.17 | 11 | *135 (71, 64, -)* |
| Huambo | -12.86 | 15.7 | 1670 | *342 (-, 250, 92)* |
| Mongu Inn | -15.26 | 23.13 | 1040 | *444 (126, 167, 151)* |
| 4STAR | - | - | - | *308 (77, 139, 92)* |

## 2.4 WRF-CAM5: Concept and Configuration

WRF-CAM5 is an adaptation of WRF-Chemistry (WRF-Chem) model (Grell et al., 2005), which integrates the physics and aerosol packages of the global CAM5 (Ma et al., 2014; Zhang et al., 2015a), making it suitable for studying multi-scale atmospheric processes and evaluating aerosol and physics parameterizations in global climate models (Wang et al., 2018). WRF-CAM5 has been widely applied to investigate air quality and climate interactions in Asia and the United States (Campbell et al., 2017; Wang et al., 2018; Zhang et al., 2015b)

The WRF-CAM5 model incorporates advanced cloud schemes, including the two-moment cloud microphysics scheme (Morrison and Gettelman, 2008), the shallow cumulus scheme of Bretherton and Park (2009), turbulence parameterization (Bretherton and Park, 2009), the Zhang-MacFarlane convective cloud scheme (Zhang and Mcfarlane, 1995), and aerosol-cloud feedback mechanisms (Song and Zhang, 2011; Lim et al., 2014). It also includes a mixed-phase and ice clouds parameterization (Niemand et al., 2012), a modal aerosol module (MAM3) (Liu et al., 2012), coupled with a gas-phase chemistry scheme (Zaveri and Peters, 1999), enabling the determination of aerosol (including smoke) properties. Aerosol optical properties are calculated using WRF-Chem routines (Fast et al., 2006) with Mie theory calculations, and cloud droplet activation based on the Fountoukis and Nenes (2005) and Zhang et al. (2015a) scheme, considering the activation of giant CCN and insoluble particles, such as dust and black carbon. The refractive indices for organic aerosols (1.45+0i) and black carbon (1.85+0.71i) were assumed to be constant across the shortwave radiation spectrum (Shinozuka et al., 2020a).

Recent studies have demonstrated the model's accuracy in capturing smoke concentration, aerosol properties, and the vertical distribution of BBA in the SEA region (Doherty et al., 2022; Chang et al., 2023). In this study, WRF-CAM5 was configured with a horizontal resolution of 36 km and 74 vertical layers across the spatial domain 41°S-14°N, 34°W-51°E, initialized every five days using the National Center for Environmental Prediction (NCEP) Final Operational Global Analysis (NCEP FNL) and Copernicus Atmosphere Monitoring Service (CAMS) reanalysis datasets as detailed in Shinozuka et al. (2020a) and Doherty et al. (2022), with daily smoke emissions from the Quick-Fire Emissions Dataset version 2 (QFED2) (Darmenov and Da Silva, 2015).



**2.5 Plume Age Derivation**

To estimate the age of the aerosols, defined as the time since emission, we utilized carbon monoxide (CO) tracers coupled with smoke emissions within the WRF-AAM model. The WRF-AAM model, which share similar configurations with WRF-CAM5, allows for region- and case-specific microphysical parameterization of aerosol-cloud interactions (Thompson and Eidhammer, 2014) and aerosol-radiation interactions (Saide et al., 2016), thus facilitating the characterization of aerosol aging processes.

The regional model was set up with a horizontal resolution of 12 km, encompassing the geographical area (41°S-14°N, 34°W-51°E). This domain size is considered sufficiently extensive to contain nearly all fires across the African continent (Saide et al., 2016; Howes et al., 2023). The fire emission source for the model is based on QFED2 which uses fire radiative power (FRP) as a proxy for estimating fire emissions. QFED2 leverages the cloud correction techniques developed in the Global Fire Assimilation System (GFAS) to refine emission estimates (Darmenov and Da Silva, 2015). Location and FRP of fires are sourced from MODIS Level 2 fire products (MOD14 and MYD14), coupled with MODIS Geolocation products (MOD03 and MYD03). The NCEP Global Forecasting System (GFS) meteorology serves as the primary driver for the WRF-AAM model which incorporates daily smoke emissions from QFED2, which are subsequently adjusted to correspond with satellite-derived AOD using a near real-time inversion algorithm, as detailed in Saide et al. (2016).

The model was run in a forecast mode to estimate the time since smoke emission, with a maximum aerosol age of 14 days. The model attaches age tracers to the CO released at BB emission sites. These tracers, which are treated as chemically inert gases, are tagged for each day, and tracked for up to two weeks, allowing sufficient time for the smoke to travel across the SEA. This model setup operates in a cycle, where each day, the tracers from the previous day are shifted to the next older tracer bin, continuing until they reach the 14-day mark. At this point, they accumulate in the oldest bin. The age of the smoke plume is determined by averaging the concentration of these tracers at any given location. However, this setup has limited accuracy at the upper end of the age estimate; for instance, smoke older than 14 days is still averaged as 14 days old, even if it contains an equally concentrated mixture of tracers that have been out for different lengths of time beyond the 14-day tracking period. In comparison, the HYSPLIT model (Stein et al., 2015), also driven by the NCEP GFS meteorology, typically yields age estimates that are approximately one day lower than the WRF-AAM. Given that HYSPLIT employs a simpler scheme at coarser resolutions, it is likely that the age estimates from WRF-AAM model are more accurate and reliable. The plume ages assigned within a given column exhibited minimal variability (see supplemental information S6), which gives confidence to the method.

To determine the effective aerosol age for AERONET and 4STAR observations, which are columnar, the WRF-AAM model's assigned plume age in the vertical layers above these instruments is weighted by their respective extinction coefficients ($\beta\_ext$) per equation (1):

$$Aerosol\ age = \frac{\int_{s_{elv}}^{toa} \beta_{ext} * plume\ age\ dz}{\int_{s_{elv}}^{toa} \beta_{ext}\ dz} \ , \qquad\qquad (1)$$



Where $s_{elv}$ is the elevation of the AERONET station or the flight altitude of observation for 4STAR and toa is the top
of the atmosphere.

## 2.6 Spatiotemporal Collocation and Analysis

The AERONET inversion products provide AOD and SSA at four wavelengths: 440, 675, 870, and 1020 nm. The
relationship between AOD and wavelength, defined by the Ångström exponent formula (Eck et al., 1999) is used to
compute AE(α):

$$AE(\alpha) = -\frac{\ln(\tau_1) - \ln(\tau_2)}{\ln(\lambda_1) - \ln(\lambda_2)},\tag{2}$$

Where $\tau_1$ and $\tau_2$ are AOD at wavelengths $\lambda_1$ and $\lambda_2$. In AERONET and 4STAR retrievals, AE(α) is calculated from
AOD measurements at 440 nm and 870 nm.

Given the differences in the temporal resolution of 4STAR and AERONET observations and the WRF-AAM model
outputs, we adapted the model output to match all AERONET and 4STAR observations spatially and temporally, to
yield the extinction-weighted aerosol age calculated using extinction at 532 nm as in Equation (1). However, since
AERONET does not provide retrievals at 532 nm, we computed the equivalent AOD and SSA at 532 nm to correspond
with the model's output using the Ångström formula (equation 2) for AOD, and the linear interpolation equation for
SSA.

## 2.7 Separating Boundary Layer (BL) contributions from Total Column (TC) observations

In the SEA, as ocean surface temperatures rise, the BL deepens and decouples, with the BL height (BLH) increasing
away from the African coast, from approximately 1300m to 1700m, before transitioning to a cumulus-dominated
cloud regime, as explained by Zhang and Zuidema (2019, 2021) and Ryoo et al. (2021). Over land, BBA dominate
the deep BL, extending beyond 6 km and are then advected over the SEA above the cloud layer by the FT winds (Ryoo
et al., 2021), particularly from August to October during strong south African Easterly Jet (AEJ-S) episodes (Adebiyi
and Zuidema, 2016). The radiative effects of aerosols in the SEA region are heavily influenced by the interplay
between the aerosol layer and cloud layer (Zhang and Zuidema, 2021; Chang and Christopher, 2017).

While BBA generally maintain their path within the FT over the SEA, large scale subsidence (Wilcox, 2010) and the
low-level easterlies (Diamond et al., 2018) occasionally bring the aerosol layer inro contact with the BL clouds,
particularly between June and August, causing the entrainment of aerosols and altering their properties (Dobracki et
al., 2024). Marine aerosols, particularly sea salt aerosols, have higher SSA values and enhanced CCN activity, which
are altered when they mix with foreign particles especially in coastal regions near anthropogenic sources (Pósfai et
al., 1995; Adachi and Buseck, 2015). Dang et al. (2022) showed that BBA sampled during ORACLES dominate the
FT while sea salt aerosols dominate the BL over the SEA with a fraction of BBA mixed with sea salt aerosols in the
BL. Therefore, our goal of investigating the evolution of BBA from TC observations is complicated by the potential
contribution of non-BBA aerosols from the MBL. To address this, and given that AERONET and 4STAR provide
columnar retrievals above the observation altitude, we employed a two-pronged approach, detailed in Section 2.7.1



and 2.7.2, to isolate the FT aerosol from the columnar observations. First, we applied a model-derived ratio to partition aerosol loading in the FT and BL over the SEA. Subsequently, we implemented a size thresholding technique to exclude contributions from larger particles, ensuring our analysis remains focused on BBA properties.

### 2.7.1 Application of model-derived extinction ratio

To address the potential influence of MBL aerosol properties on the TC observations by AERONET and 4STAR, we use a model-derived vertical distribution of extinction to estimate the FT contributions to the TC measurements. Specifically, we applied model-derived ratios of extinction in the BL relative to the TC ($R_{BL}$), and in the FT relative to the TC ($R_{FT}$) to:

(i)   all AERONET observation at Ascension Island and Namibe;

(ii)  4STAR observations when the NASA P-3 aircraft was flying within the BL.

To compute $R_{BL}$ (equation 3) and $R_{FT}$, we applied BLH values from WRF-CAM5 for both sets of observations. The BLH values are estimated according to (Chang et al., 2023) and validated with radiosonde observations at Ascension Island (Zhang and Zuidema, 2019).

$$R_{BL} = \frac{\int_{S_{elv}}^{BLH} \beta_{ext} dz}{\int_{S_{elv}}^{toa} \beta_{ext} dz},\tag{3}$$

$$R_{BL} = 1 - R_{BL},\tag{4}$$

Using the computed ratios, we partitioned the columnar AOD ($AOD_{TC}$) measurements at locations (i) and (ii)using the following equations:

$$AOD_{BL} = R_{BL} * AOD_{TC},\tag{5}$$

$$AOD_{FT} = R_{FT} * AOD_{TC},\tag{6}$$

Here, $AOD_{BL}$ and $AOD_{FT}$ represent the partial AOD at 532 nm within the BL and FT, respectively. For (i), $AOD_{TC}$ refers to the columnar AOD above the site's elevation and $AOD_{BL}$ represents the portion of AOD between that elevation and the BLH. For (ii), $AOD_{TC}$ refers to the columnar AOD above the P-3 aircraft altitude, while $AOD_{BL}$ represents the portion of AOD between the P-3 flight altitude and the BLH.

We next calculated the proportion of aerosol age and SSA within the FT using the following:

$$Aerosol_{ag\ FT} = \frac{\int_{BLH}^{toa} \beta_{ext} * plume\ age\ dz}{\int_{BLH}^{toa} \beta_{ext}\ dz},\tag{7}$$

$$SSA_{FT} = \frac{(SSA_{TC} * AO_{TC}) - (SSA_{BL} * AOD_{BL})}{AOD_{FT}},\tag{8}$$

While the WRF-CAM5 BLH values generally agree with observational data, our analysis indicates that the model significantly underestimates aerosol loading in the BL, a limitation also highlighted in prior studies (Figure 8-11,



Doherty et al. (2022)). Despite having the most accurate model representation of aerosol properties in the SEA region (Doherty et al., 2022), this underestimation in the BL by WRF-CAM5 (see supplemental Figures S7 – S9) affects the representation of BBA properties in the FT at location (i) and (ii) , with physically unrealistic values. To address this, we applied an assumed $SSA_{BL}$ value of 1, recognizing that this introduces a source of uncertainty to the analysis, as it assumes that all aerosols in the BL are purely scattering and does not account for the occasional mixing of BBA with

marine sea salt in the BL. To aid the separation of BL and FT BBA, we applied an AE screening, described below.

### 2.7.2 Application of Ångstrom Exponent (AE) Thresholds

The extinction Ångstrom Exponent (AE) is sensitive to the size distribution of aerosol particles (Eck et al., 1999; Schuster et al., 2006; Aladodo et al., 2022). Fine-mode aerosol particles, such as smoke typically have a higher AE (usually > 1) than coarse-mode aerosols such as marine sea salt, which typically have a lower AE (< 1). Along with

implementing the model-derived extinction ratio outlined in Section 2.7.1, to further discard contributions of other aerosols to the columnar measurements, particularly in atmospheric layers above location (i) and (ii), we applied an AE filter. We posit that AE values < 1 would indicate the presence of sea salt aerosol whereas AE values ≥ 1.5 would indicate the dominance of BBA (Eck et al., 1999) while layers with measured AE values between 1 and 1.5 contain varying degrees of sea salt and BBA following the results of Dang et al. (2022). We tested four AE threshold values:

≥ 0.75, ≥ 1, ≥ 1.2, and ≥ 1.4, to successively exclude such measurements, retaining only fine particles while carefully considering the data processing. Our analysis of the AE thresholding (see supplemental information S3 for more details) suggests that AE of 1.5 for BBA might be applicable near emission sources, lower values during transport and aging led us to choose a lower threshold of 1.2 to have more robust statistics over the ocean.

### 3 Results and Discussion

### 3.1 Age distribution of smoke plumes

The aerosol age (a) and extinction (b) on October 17, 2018, as derived by the WRF-AAM model is shown in Figure 3, with the altitude track of the P-3 aircraft, which flew over the SEA between 7:00 – 15:00 UTC (see also supplemental Figure S10), overlaid onto the curtain plot. For this case, the model simulates particles at altitudes below 2 km and above 8 km to be substantially older, with plume age ranging from 9 – 12 days, than those in the mid FT (3

– 7 km) with age ranging from 3 – 8 days. The age of the plume sampled between 12:00 – 13:00 UTC at the flight intersection with the HYSPLIT trajectory shown in Fig. 1 ranged from 4 – 8 days, relative to HYSPLIT age range of 4 – 6 days (see supplemental Figure – S1). In Figure 3(b), the vertical distribution of aerosol extinction generally revealed maximum extinction within the lowest 2km. The model demonstrated a consistent inverse relationship between extinction and height, with extinction decreasing as height increased, reaching a minimum above 6 km.

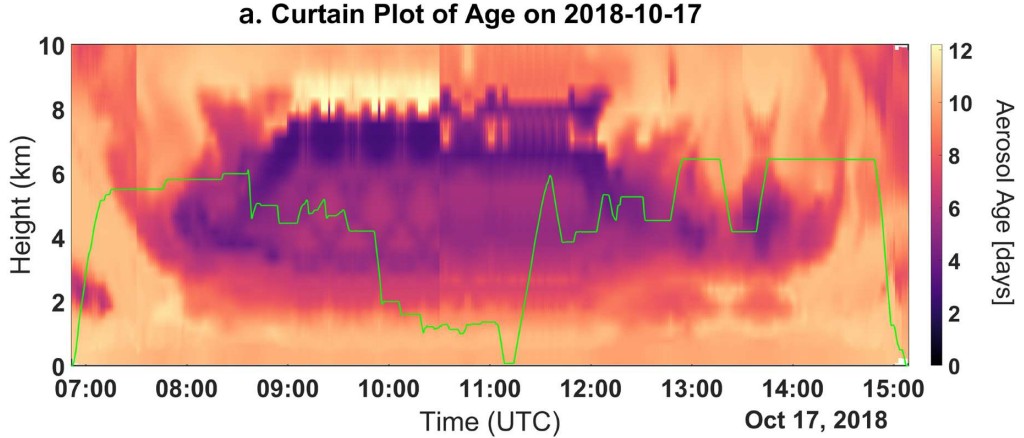

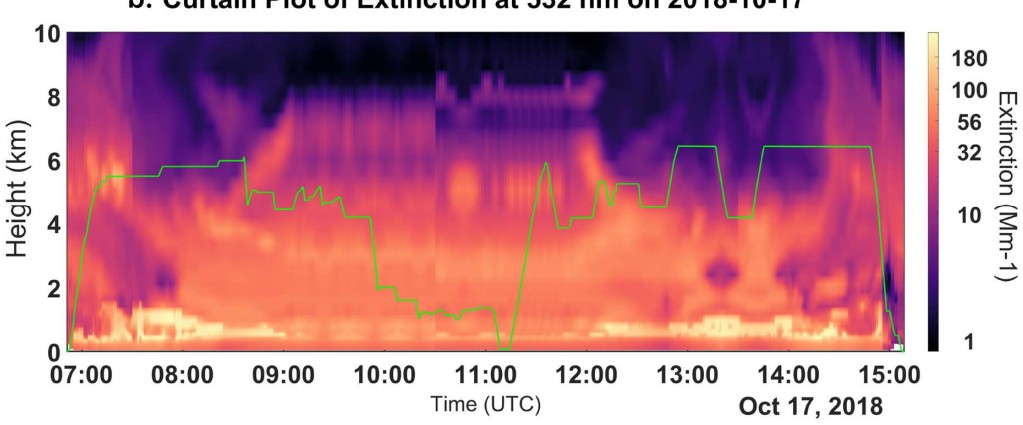


**Figure 3: WRF-AAM curtain plot with altitude on the Y-axis and time in UTC on the X-axis, showing the smoke plume age forecast (a) and aerosol extinction (b) along the P-3 track (solid green line) during research flight 10 of the third ORACLES deployment on October 17, 2018. The smoke plume age is calculated as the average of the tracer concentration in each age bin.**

The modelled vertical distribution of aerosol age and extinction across all ORACLES flight missions and exemplified in Fig. 3 reveals a distinct pattern: aerosols around 7km, within a central altitude band of 3 – 8 km, are younger and likely represent aerosols recently transported by the FT jet. In contrast, aerosols below 1 km and above 8 km over the ocean appear to have an older connection to their source regions, suggesting they either followed a different transport route than the FT aerosols or were recirculated by the AEJ-S, as previously described by Adebiyi

and Zuidema (2016).



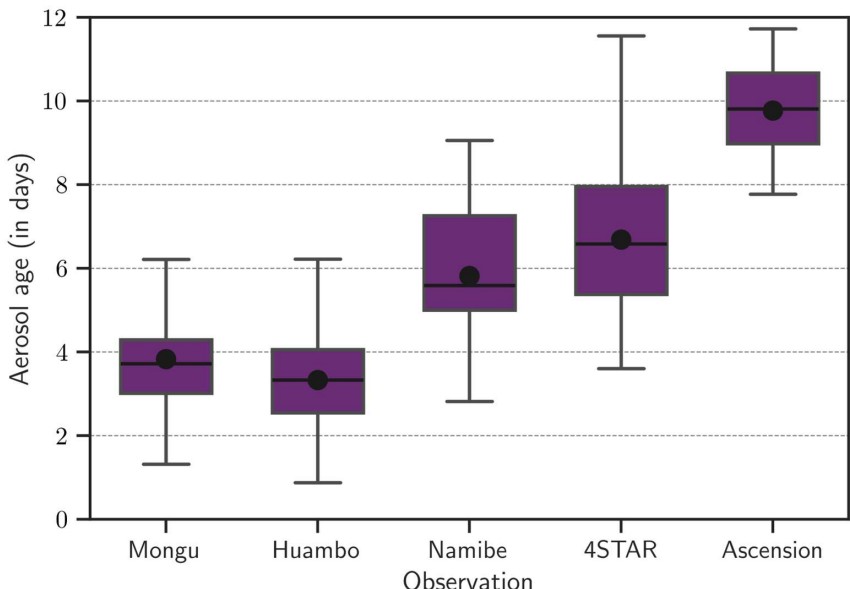

**Figure 4: Box-whisker plots of the extinction-weighted aerosol age for available observation at each site. The boxes represent the 25th (lower box) and 75th (upper box) percentile while the whiskers show the 10th (lower whisker) and 90th (upper whisker) percentiles of aerosol age. The black circle represents the mean aerosol age while the solid line across the box represents the median. Data plotted represents the columnar measurements across all observations.**

The summary statistics of aerosol age from all AERONET and 4STAR observations considered in this study (Fig. 4) show the oldest BBA are observed near Ascension Island (7.9°S, 14.4°W), a remote island about 2000 km away from the African coast. WRF-AAM outputs predict the samples at Ascension Island to be aged between approximately 7 – 12 days, with a mean age of approximately 10 days, while for 4STAR observations, WRF-AAM show that the aerosol age ranged between 3 and 12 days, with a mean age of about 7 days. However, younger BBA were observed over land, closer to the source region of the fires. The plumes sampled at Mongu, Huambo, and Namibe, were aged between 1 – 7 days, 0 – 6 days, and 3 – 9 days, respectively, with mean age of approximately 4, 3.5, and 6 days. The youngest aerosols in the source region were sampled at Huambo (12.8°S, 15.7°E) and Mongu (15.2°S, 23.1°E), with over 75% of all observations at both stations aged less than 4 days. The mean age of aerosols near the African coast was approximately 6 days, suggesting that the smoke measured at Namibe (15.15°S, 12.17°E) has migrated from the primary burning region at Mongu and Huambo. This is further supported by trajectory analysis (see supplemental Figure S1) and by the limited count of active fires recorded in the vicinity of Namibe during the campaign period as shown in Fig. 1. 4STAR observations show a wider age range than those from AERONET sites. This variability is likely attributed to the ORACLES flight strategy, which sometimes targeted freshly emitted smoke from new fires, resulting in the lower age range of 3 to 6 days. The modelled aging pattern suggests the smoke originating from the continent follows a relatively steady trajectory up to Ascension Island. This is supported by HYSPLIT trajectory analysis (see supplementary information – S2), which reveals that the majority of air masses from the burning region follow a similar path westward, typically reaching Ascension Island within 10 to 12 days.



## 3.2 Changes in total column SSA with age

Next, we investigate the relationship between SSA and the model-derived age, stratified by the AE in the TC over the entire study region. We do this by combining the available collocated AERONET, 4STAR and WRF-AAM output datasets from September 2016, August 2017, and October 2018. Combining data from all three months together provides additional dynamic of change in SSA with time through the burning season months as shown in Eck et al. (2013). Figure 5 shows SSA at 532 nm and AE as a function of model-derived aerosol age for all observations in the

TC. Their distribution by observation site is shown in Fig. 6. The cluster of high-AE data, seen between 0 to 4 days in Fig. 5, is primarily made up of measurements in the burning region from Mongu and Huambo, with average AE values of 1.81 and 1.86, respectively (Fig. 6), indicating a dominance of BBA in the TC at these locations. Correspondingly, the TC SSA ranges between 0.77 and 0.97, with mean TC SSA of 0.86 at the two locations, with over 75% of the data below 0.90.

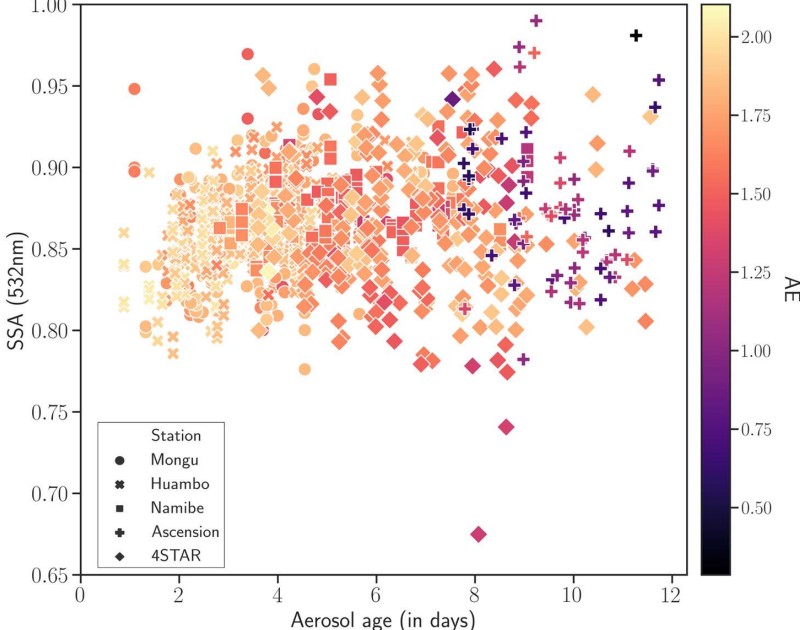

**Figure 5: Relationship between SSA$_{532nm}$ (y-axis), AE (color bar), and aerosol age (x-axis) in the total atmospheric column (TC). The different markers represent the site of observation while the marker shading represents the AE.**





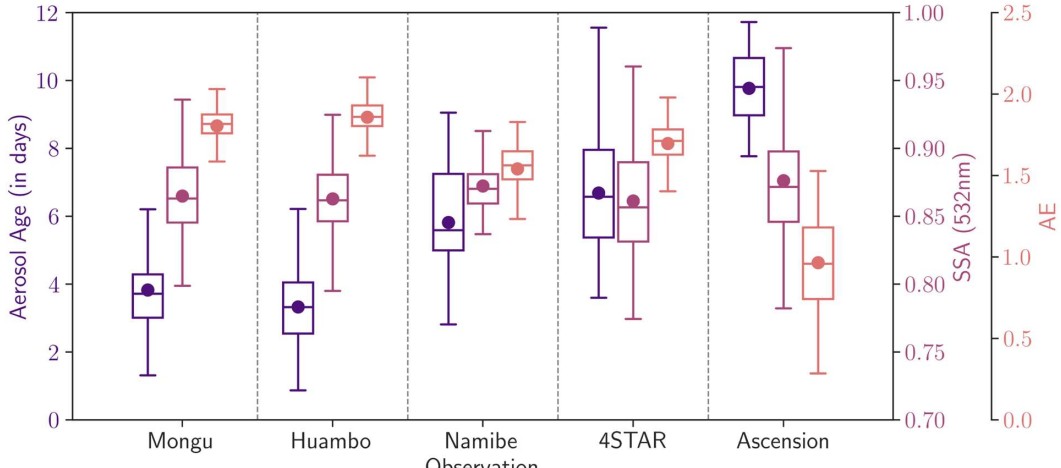

**Figure 6: Box and whisker plots showing the distribution of aerosol age (violet), SSA$_{532nm}$ (magenta) and AE (peach) in the TC for available observation at each site. The box-whisker plot shows the 10th (lower whisker), 25th (lower box), 50th (median), 75th (upper box), and 90th (upper whisker) percentiles. The mean value for each parameter is marked with the circle.**

The TC SSA result in the source region at Mongu and Huambo implies that fresh BBA ($\leq$ 4 days) is optically dark and highly absorbing of solar radiation as indicated by low SSA values (Table 4), which is consistent with previous findings (Haywood et al., 2003; Abel et al., 2003; Leahy et al., 2007; Eck et al., 2013) in the region. A decrease in TC AE is observed with increasing distance from the source region (see supplemental Figure S14), indicating larger particle sizes at more remote locations over the SEA. At Namibe, the mean TC AE is 1.54, with a mean TC SSA of 0.87, ranging from 0.83 to 0.95. This observed trend of increasing particles size align with in-situ aerosol size measurements (Howes et al., 2023). Aerosols over the SEA exhibit a broad range of TC AE values, from 0.28 and 2.0, reflecting significant heterogeneity in particle sizes within the TC. These observations have a mean TC SSA of 0.86 for 4STAR, with a range between 0.67 – 0.96, and a mean TC SSA of 0.87 at Ascension Island, ranging from 0.78 to 0.99. Some retrievals over the SEA showed both low SSA values ($< 0.8$) and low AE values ($\leq 1.0$), suggesting the possible presence of other aerosol types at Ascension Island (Howes et al., 2023), or a combination of coarse non- or less-absorbing particles and BBA within the same column.

Upon applying an AE threshold of 1.2 to the TC, we binned the aerosol age into two-day intervals and estimated the mean TC SSA for each bin (Fig. 7). The result shows that mean TC SSA increases with age, changing from 0.84 within 0 to 2 days of emission over land in the source region up to 0.87 after 6 to 8 days, then decreases over the ocean, reaching approximately 0.86 after 10 to12 days. This result indicates changes in aerosol optical properties during different stages of transport.





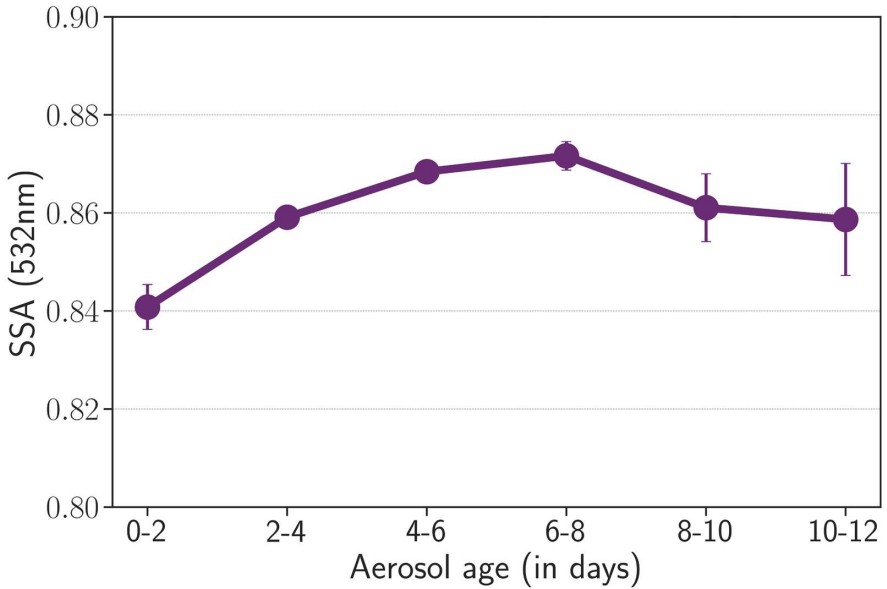


**Figure 7: Mean SSA (at 532 nm) within the vertical column (TC SSA) as a function of aerosol age for the optimal threshold (AE ≥ 1.2). Error bars represent the standard error of the mean.**

### 3.3 Vertical partitioning of aerosol properties.

In this section, we present and analyze the results of the vertical partitioning of aerosol properties, utilizing the model-derived extinction ratio as outlined in Section 2.7.1. Through this method, we made comparisons between the optical properties of aerosols residing in the FT and those in the TC, providing insights into the distribution of aerosols across the two layers.

### 3.3.1 Free Tropospheric Aerosol Loading

          The fraction of AOD in the free troposphere (FT AOD) relative to the total column AOD (TC AOD) over
the SEA, derived from the WRF-CAM5 model, is shown in Figure 8. For AERONET, TC AOD represents the columnar AOD above the site elevation, while for 4STAR, TC AOD refers to the AOD above the aircraft altitude. The modelled FT AOD fraction exhibits considerable spatial variation, with contributions to TC AOD ranging from 0.42 to 0.92 (mean: 0.73) at Ascension Island, 0.68 to 0.99 (mean: 0.95) for 4STAR observations, and 0.70 to 0.95 (mean: 0.86) at Namibe. Only 4STAR sky scans conducted at flight altitudes below the BL top were included in the
separation, representing approximately 48% of the total 4STAR sky scans. The differences in the fraction of FT AOD between Ascension Island, 4STAR, and Namibe (as shown in Fig. 8) can be attributed primarily to the relative proximity to the source regions. The 4STAR measurements, taken predominantly east of Ascension Island, captured denser smoke plumes before they reached the island. Additionally, the higher FT AOD in 4STAR compared to AERONET is likely due to unaccounted AOD between the aircraft altitude and the surface in the 4STAR dataset. This
finding emphasizes that a substantial portion of TC AOD, exceeding 50% on average resides in the FT, indicating a





higher aerosol loading in the FT compared to the MBL over the SEA. Subsequently, the model-derived fraction (Equation 4) is applied to estimate FT AOD for Ascension Island, 4STAR and Namibe retrievals using Equation (6). At Ascension Island, the mean FT AOD for the campaign period is approximately 0.23 compared to the mean TC AOD of 0.31, as shown in Figure 9 and summarized in Table 3. However, higher mean FT AOD values of 0.35 and

0.76 are observed from 4STAR and at Namibe respectively. The FT AOD from 4STAR aligns with findings of Shinozuka et al. (2020b) who reported that daytime AOD above-cloud is similar to that in clear-sky conditions over the SEA. The observed FT aerosol loading from AERONET and 4STAR measurements over the SEA ocean agrees with the modelled FT AOD reported in Chang et al. (2023) for the same region.

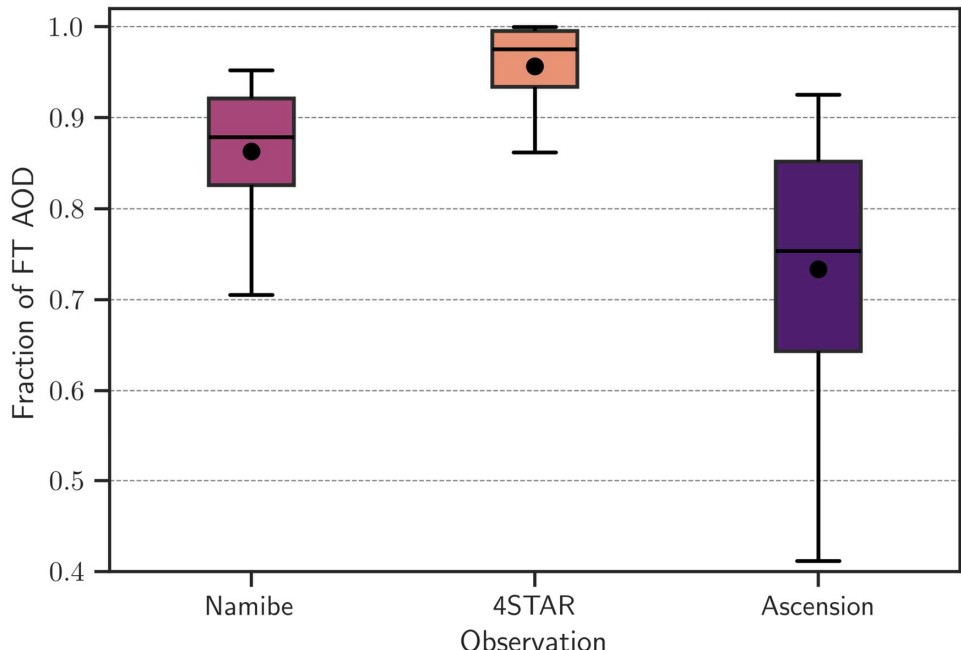

**Figure 8: Model-derived fraction of free tropospheric (FT) AOD to total column (TC) AOD. A fraction of 1 means that aerosol loading is completely in the FT. The box-whisker plot shows the 10th (lower whisker), 25th (lower box), 50th (median), 75th (upper box), and 90th (upper whisker) percentiles of fraction of FT AOD. The black circle represents the mean fraction. Note: the outliers have been hidden in the figure. Therefore, on average, more than 50% of the aerosol loading in the ORACLES observation region is in the FT.**




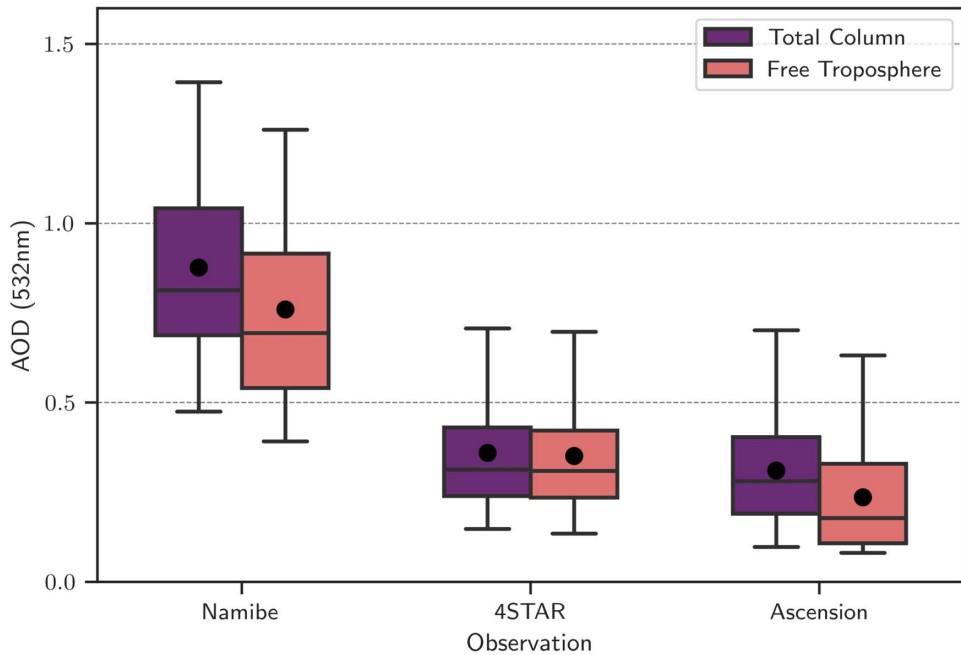

**Figure 9: Comparison of AOD (at 532 nm) in the total column (TC) and free troposphere (FT) over the southeast Atlantic (SEA) ocean. The box-whisker plot shows the 10th (lower whisker), 25th (lower box), 50th (median), 75th (upper box), and 90th (upper whisker) percentiles. Solid black circles represent mean AOD values.**




**Table 3: Summary statistics of (a) Fraction of FT AOD (b) TC AOD and FT AOD (c) TC AE and FT AE**

| (a) Fraction of FT AOD | | | | | | | | |
|---|---|---|---|---|---|---|---|---|
| Site Name | Count | Mean | Std | Min | 25% | 50% | 75% | Max |
| Ascension | 69 | 0.73 | 0.14 | 0.42 | 0.64 | 0.75 | 0.85 | 0.92 |
| Namibe | 133 | 0.86 | 0.07 | 0.70 | 0.82 | 0.88 | 0.92 | 0.95 |
| 4STAR | 122 | 0.95 | 0.05 | 0.68 | 0.93 | 0.97 | 0.99 | 0.99 |

| (b) TC AOD (FT AOD) 532 nm | | | | | | | | |
|---|---|---|---|---|---|---|---|---|
| Site Name | Count | Mean | Std | Min | 25% | 50% | 75% | Max |
| Ascension | 69 | 0.31 | 0.15 | 0.10 | 0.18 | 0.28 | 0.40 | 0.88 |
| | | (0.23) | (0.15) | (0.08) | (0.10) | (0.18) | (0.33) | (0.81) |
| Namibe | 133 | 0.88 | 0.28 | 0.47 | 0.69 | 0.81 | 1.04 | 1.74 |
| | | (0.76) | (0.26) | (0.39) | (0.54) | (0.69) | (0.91) | (1.58) |
| 4STAR | 255 | 0.36 | 0.15 | 0.14 | 0.24 | 0.31 | 0.43 | 0.85 |
| | | (0.35) | (0.15) | (0.13) | (0.23) | (0.30) | (0.42) | (0.84) |

| (c) TC AE (FT AE) 440-870 nm | | | | | | | | |
|---|---|---|---|---|---|---|---|---|
| Site Name | Count | Mean | Std | Min | 25% | 50% | 75% | Max |
| Ascension | 69 | 0.96 | 0.27 | 0.28 | 0.74 | 0.96 | 1.18 | 1.52 |
| | | (0.97) | (0.28) | (0.28) | (0.75) | (0.97) | (1.20) | (1.54) |
| Namibe | 133 | 1.54 | 0.15 | 1.01 | 1.47 | 1.56 | 1.64 | 1.82 |
| | | (1.55) | (0.15) | (1.01) | (1.47) | (1.57) | (1.65) | (1.83) |
| 4STAR | 255 | 1.69 | 0.15 | 0.86 | 1.63 | 1.71 | 1.78 | 2.08 |
| | | (1.70) | (0.15) | (0.86) | (1.63) | (1.71) | (1.78) | (2.08) |




### 3.3.2 Free Tropospheric Ångstrom Exponent

The FT AE values observed from 4STAR range between approximately 1.0 and 2.1, averaging 1.7, consistent with findings from Leblanc et al. (2020), that reported minimal spatial dependence of above-cloud AE in the SEA. At

Namibe, FT AE ranges from 1.0 – 1.82 with mean of about 1.55.

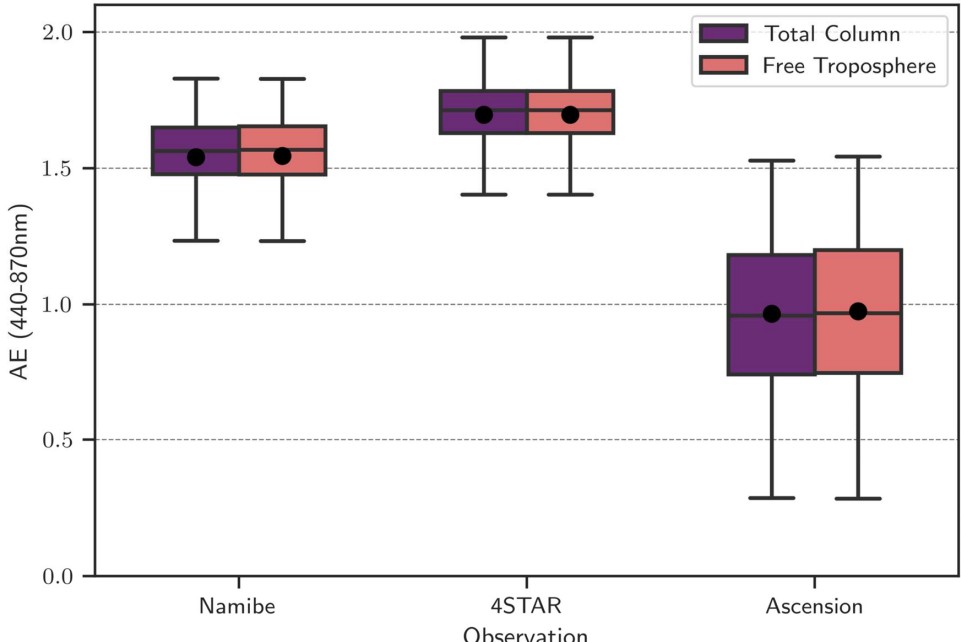

**Figure 10: Same as Figure 8 but for AE (440-870 nm)**

At Ascension Island, the FT AE ranges from 0.28 to 1.5 with a mean of approximately 1.0, indicating a variety of

particle sizes. The AE distribution within the FT mirrors the AE values obtained from the TC, suggesting that the particles' intensive properties remain largely unchanged after isolating MBL contributions. Typically, TC AOD at Ascension that includes the other aerosol contribution would be expected to result in a lower AE value than the FT AOD-derived value of AE due to the presence of sea salt aerosols and dimethyl sulfide (DMS) in the maritime environment (Smirnov et al., 2002). However, the lower FT AE values at Ascension Island suggest there is a source

of advected coarse particles in the FT, especially for cases with FT AE as low as 0.5 shown in Fig. 10. The FT AE values from 4STAR in the ORACLES campaign region primarily reflect the characteristics of BBA, with mean AE values greater than 1.2 (Table 3), consistent with previous studies in the region (Pistone et al., 2019; Leblanc et al., 2020).



### 3.3.3 Free Tropospheric Single Scattering Albedo

Figure 11 shows the distribution of SSA, AE and aerosol age in the free troposphere by observation site. Over the SEA, the FT SSA differs from that of the TC SSA (shown in Figure 6). At Namibe, the mean FT SSA is 0.85, from a range of 0.79 – 0.94. 4STAR observations show FT SSA values between 0.67 - 0.96, with a mean of 0.86. At Ascension Island, the FT SSA ranged from 0.60 - 0.98, with mean of 0.82 (Table 4). These values of FT SSA are lower than TC SSA due to exclusion of the BL portion of the TC, which may include other non-BBA aerosols with

higher SSA. This difference between FT SSA and TC SSA over the SEA is further discussed in section 3.4. Given that there was no vertical partitioning in the source region since BBA dominates the full column, the FT SSA and TC SSA are equal.

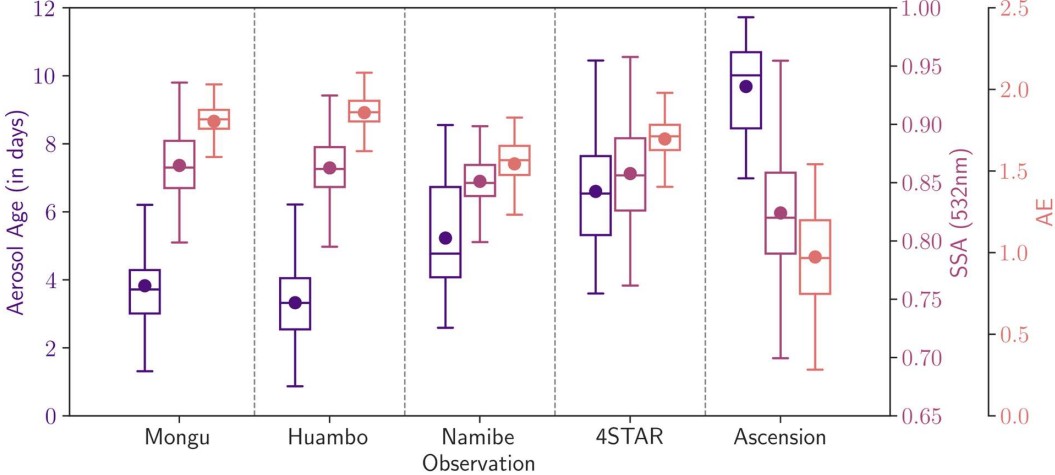

**Figure 11: Same as in Figure 6, but for the free troposphere (FT).**




**Table 4: Summary statistics of TC and FT (in parentheses) aerosol age (a), SSA (b) and AE (c) based on observations site**

| a. Result Age | | | | | | | | |
|---|---|---|---|---|---|---|---|---|
| **Site Name** | **Count** | **Mean** | **Std** | **Min** | **25%** | **50%** | **75%** | **Max** |
| Ascension | 69 | 9.77 | 1.16 | 7.77 | 8.99 | 9.97 | 10.7 | 11.73 |
| | | (9.68) | (1.32) | (6.98) | (8.45) | (10.02) | (10.69) | (11.73) |
| Namibe | 133 | 5.82 | 1.49 | 2.82 | 5.00 | 5.59 | 7.25 | 9.05 |
| | | (5.23) | (1.55) | (2.59) | (4.08) | (4.78) | (6.73) | (8.55) |
| Huambo | 336 | 3.33 | 1.10 | 0.87 | 2.54 | 3.32 | 4.05 | 6.80 |
| Mongu | 355 | 3.83 | 1.25 | 1.09 | 3.01 | 3.72 | 4.29 | 7.55 |
| 4STAR | 255 | 6.69 | 1.76 | 3.60 | 5.37 | 6.58 | 7.96 | 11.56 |
| | | (6.60) | (1.72) | (3.60) | (5.32) | (6.54) | (7.64) | (11.78) |

| b. Single Scattering Albedo (SSA 532 nm) | | | | | | | | |
|---|---|---|---|---|---|---|---|---|
| **Site Name** | **Count** | **Mean** | **Std** | **Min** | **25%** | **50%** | **75%** | **Max** |
| Ascension | 69 | 0.87 | 0.04 | 0.78 | 0.85 | 0.87 | 0.90 | 0.99 |
| | | (0.82) | (0.07) | (0.60) | (0.79) | (0.82) | (0.86) | (0.98) |
| Namibe | 133 | 0.87 | 0.02 | 0.83 | 0.86 | 0.87 | 0.88 | 0.95 |
| | | (0.85) | (0.02) | (0.79) | (0.84) | (0.85) | (0.86) | (0.94) |
| Huambo | 336 | 0.86 | 0.02 | 0.78 | 0.85 | 0.86 | 0.88 | 0.92 |
| Mongu | 355 | 0.86 | 0.03 | 0.77 | 0.84 | 0.86 | 0.89 | 0.97 |
| 4STAR | 255 | 0.86 | 0.04 | 0.67 | 0.83 | 0.85 | 0.89 | 0.96 |
| | | (0.86) | (0.05) | (0.67) | (0.82) | (0.85) | (0.88) | (0.96) |

| c. Extinction Angstrom Exponent (AE 440-870 nm) | | | | | | | | |
|---|---|---|---|---|---|---|---|---|
| **Site Name** | **Count** | **Mean** | **Std** | **Min** | **25%** | **50%** | **75%** | **Max** |
| Ascension | 69 | 0.96 | 0.27 | 0.28 | 0.74 | 0.96 | 1.18 | 1.52 |
| | | (0.97) | (0.28) | (0.28) | (0.75) | (0.97) | (1.20) | (1.54) |
| Namibe | 133 | 1.54 | 0.15 | 1.01 | 1.47 | 1.56 | 1.64 | 1.82 |
| | | (1.55) | (0.15) | (1.01) | (1.47) | (1.57) | (1.65) | (1.83) |
| Huambo | 336 | 1.86 | 0.11 | 0.95 | 1.80 | 1.86 | 1.93 | 2.10 |
| Mongu | 355 | 1.81 | 0.11 | 1.33 | 1.76 | 1.81 | 1.87 | 2.05 |
| 4STAR | 255 | 1.69 | 0.15 | 0.86 | 1.63 | 1.71 | 1.78 | 2.08 |
| | | (1.70) | (0.15) | (0.86) | (1.63) | (1.71) | (1.78) | (2.08) |



### 3.4 Evolution of BBA SSA with age in the free troposphere

To filter out larger particles, likely non-BBA within the free troposphere, particularly over Ascension Island, we applied a threshold of AE ≥ 1.2 to the FT dataset, allowing us to focus our analysis on the temporal evolution of FT SSA during BBA-laden episodes. Our findings in Fig. 12 show a progressive increase in mean FT SSA from 0.84 to 0.87 as the aerosols age from 0 to 8 days. After 8 days, FT SSA decreases from 0.87, reaching 0.84 after approximately 12 days of transport. This trend in the mean FT SSA across age bins, summarized in Table 5, begins at 0.84 (0-2 days), increases to 0.857 (2-4 days), 0.862 (4-6 days), and peaks at 0.871 (6-8 days), before declining to 0.84 (8-10 days) and 0.845 (10-12 days). However, it is important to note the higher uncertainty in FT SSA for the 10-12 day age bin due to the reduced sample size following the AE filtering (Table 5). We analyzed the contribution of FT SSA from each campaign month to the age bins after filtering (see Table S1 and S2).

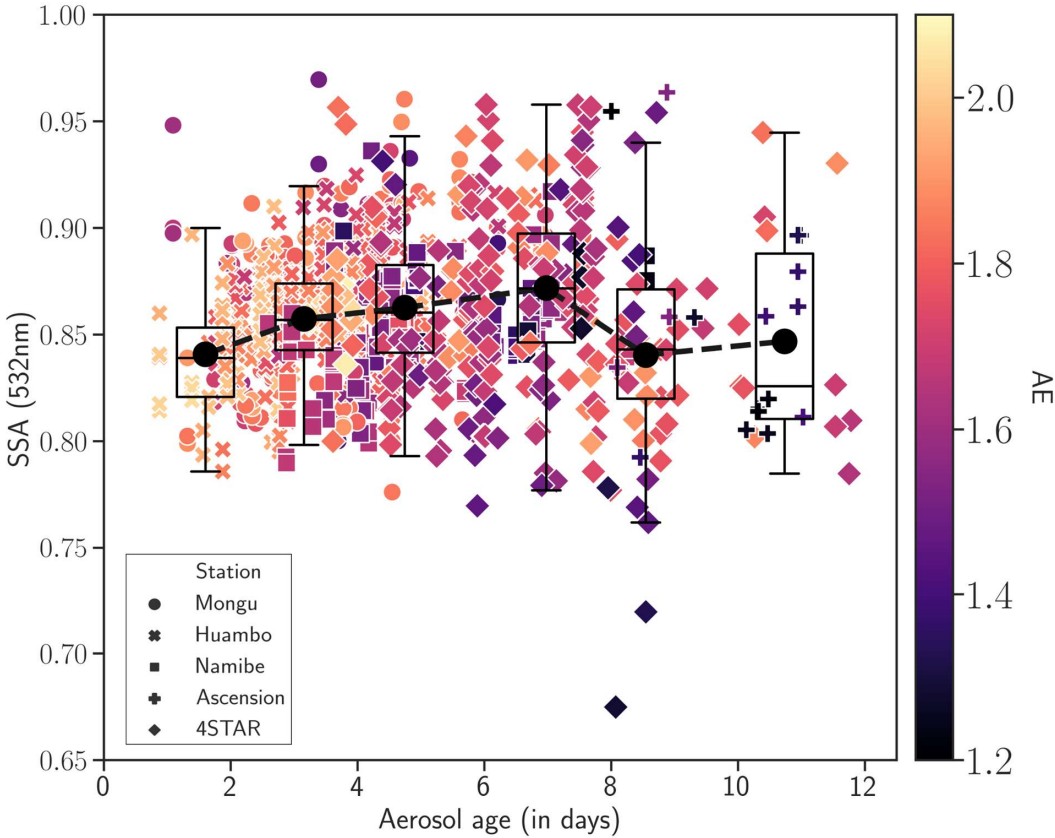

**Figure 12: Evolution of FT SSA as a function of aerosol age for the optimal threshold (AE ≥ 1.2). Scatterplot shows the relationship between SSA, AE and aerosol age for each observation. The box-whisker plots illustrate the distribution of SSA for the categorized aerosol age bins: [0-2], [2-4], [4-6], [6-8], [8-10], [10-12], showing the 10th (lower whisker), 25th (lower box), 50th (median), 75th (upper box), and 90th (upper whisker) percentiles. The mean FT SSA for each age bin is represented in the black circles.**





The analysis showed the following: no October (2018) retrievals are present in the 10-12 day age bin, and only a few are found in the 8-10 day bin; the 8-10 day bin contains a relatively equal distribution of retrievals between August and September; and in the 10-12 day bin, the ratio of September to August retrievals is approximately 3:1. These distributions show that the observed decrease in SSA within the 8-10 and 10-12 day age bins, compared to younger

plume ages, is not driven by overrepresentation of August data, earlier in the burning season, for which prior studies have identified climatologically lower SSA in the region. This is significant, as it implies that other factors, rather than the timing of the emission, drive the observed SSA change in aged plumes. The maximum value of mean FT SSA at 6-8 days shows that the BBA absorptivity decreases as the aerosol ages from emission to about 6-8 days. After 6-8 days, the FT SSA decreases, indicating that older BBA are more efficient at absorbing solar radiation. This evolution

of FT SSA supports the observations from previous studies where it was reported that the SSA initially increases within a few hours of emission and peaks on the fourth day before decreasing (Sedlacek et al., 2022). The results presented here also support the findings from Dobracki et al. (2023) who showed that in the ORACLES region, SSA decreases as the aerosol ages because of a reduction in organic aerosol mass concentrations.

**Table 5: Summary statistics of TC SSA and FT SSA (in parentheses) for each age group at (AE ≥ 1.2)**

| Age Group | Count | Mean | Std | Min | 25% | 50% | 75% | Max | SE |
|---|---|---|---|---|---|---|---|---|---|
| 0-2 | 47 | 0.841 | 0.03 | 0.785 | 0.820 | 0.840 | 0.853 | 0.948 | 0.004 |
| | (47) | (0.84) | (0.03) | (0.785) | (0.820) | (0.840) | (0.853) | (0.948) | (0.004) |
| 2-4 | 425 | 0.859 | 0.02 | 0.795 | 0.845 | 0.859 | 0.874 | 0.960 | 0.001 |
| | (430) | (0.857) | (0.03) | (0.790) | (0.843) | (0.857) | (0.874) | (0.96) | (0.001) |
| 4-6 | 319 | 0.868 | 0.03 | 0.776 | 0.847 | 0.865 | 0.887 | 0.960 | 0.001 |
| | (327) | (0.862) | (0.03) | (0.769) | (0.841) | (0.860) | (0.882) | (0.960) | (0.001) |
| 6-8 | 169 | 0.871 | 0.04 | 0.778 | 0.847 | 0.873 | 0.895 | 0.958 | 0.003 |
| | (167) | (0.871) | (0.04) | (0.777) | (0.846) | (0.871) | (0.897) | (0.958) | (0.003) |
| 8-10 | 64 | 0.861 | 0.05 | 0.675 | 0.829 | 0.860 | 0.896 | 0.970 | 0.007 |
| | (49) | (0.840) | (0.05) | (0.675) | (0.820) | (0.843) | (0.871) | (0.963) | (0.008) |
| 10-12 | 16 | 0.858 | 0.04 | 0.802 | 0.827 | 0.843 | 0.901 | 0.944 | 0.011 |
| | (23) | (0.845) | (0.04) | (0.785) | (0.810) | (0.825) | (0.890) | (0.944) | (0.009) |


We examined the mean FT SSA against the mean TC SSA at the established optimal AE threshold (Fig. 13) to gain a clearer understanding of the evolution of SSA shown in Fig. 12 and to highlight the significance of employing the combination of the model-based vertical extinction separation with an AE filter. The results demonstrate that FT SSA decreases by more than 2% for BBA aged beyond 8 days. This decrease in mean FT SSA compared to TC SSA

is consistent with the desired isolation of fine, absorbing BBA in the FT and suggests that applying only an AE filter to the TC observations isolates the contribution of larger particles. However, the AE filter does not fully isolate less-absorbing fine particles, which could include BBA mixed with scattering marine aerosols within the boundary layer, as shown by Dang et al. (2022), or non-absorbing fine-mode marine aerosols (Fitzgerald, 1991). Overall, the combined



use of both techniques, reveals a distinct evolution pattern of BBA in the FT. To examine the sensitivity of our
assumption that $SSA_{BL} = 1$, we tested alternative $SSA_{BL}$ values to account for varying degrees of absorption within
the BL. The results showed an evolution of FT SSA similar to that in Figure 13, with a decrease in mean FT SSA after
8 days (supplemental Figure S6).

The changes in SSA presented in this study are primarily associated with chemical and physical processes in
the atmosphere (Dobracki et al., 2023). Fresh aerosols over the continent show a low SSA, likely a result of a high
proportion of rBC from flaming fires. As these aerosols age in the atmosphere, they accumulate organic coatings, a
process that concurrently occurs with homogeneous nucleation of secondary organic aerosol (SOA) from volatile
organic compounds (VOCs). These processes happen rapidly within hours and continue for the first few days,
increasing the contribution of OA to the aging particles, thereby increasing SSA. After 6-8 days, the SSA starts to
decrease, possibly because SOA production slows down and heterogeneous oxidation repartitions some of the aerosol
mass back into the gas phase. In addition to changes in BBA composition from accumulation and/or evaporation of
organic coatings, there is likely some concurrent change in fine mode particle size that occurs in these aging processes.
Fine mode scattering efficiency deceases as fine mode particle size decreases and this also can result in lower SSA for
the smaller particles.

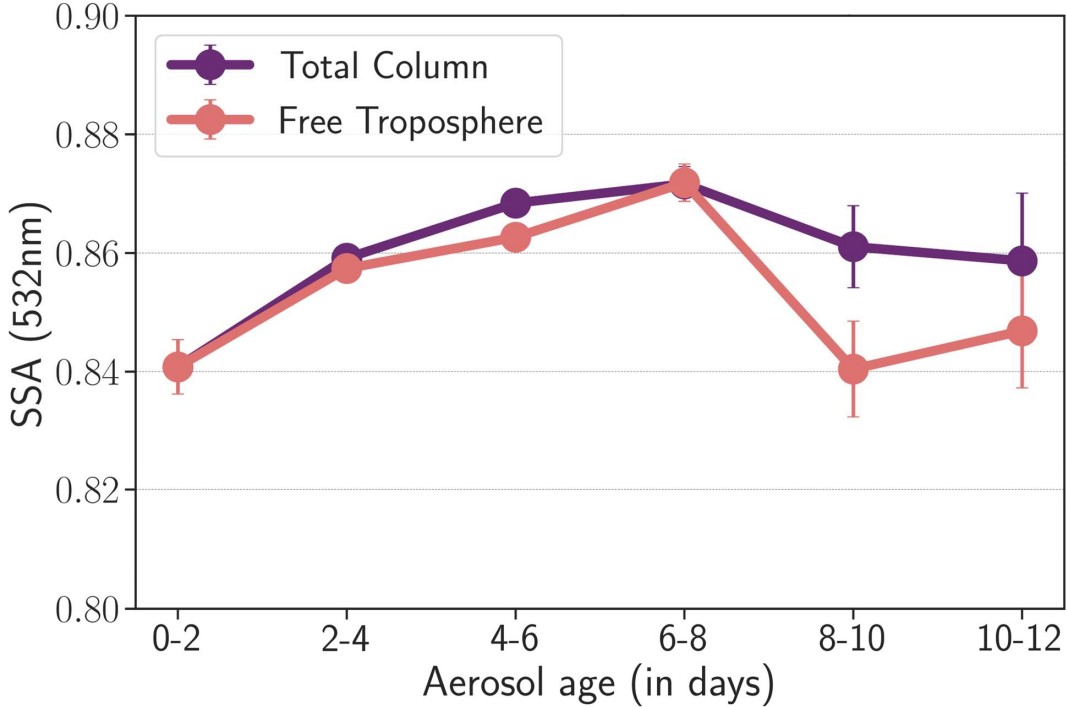

**Figure 13: Comparison between evolution of TC SSA and evolution of FT SSA for all the aerosol age bin at the optimal
threshold (AE ≥ 1.2). Error bars represent the standard error of the mean.**



## 4 Conclusions

We introduce a novel approach to investigate the evolution of optical properties of BBA during long-range transport from continental southern Africa to the SEA region. This approach integrates ground-based and airborne remote sensing measurements from AERONET and 4STAR, collected during the three ORACLES deployments, with model output from two WRF configurations: WRF-CAM5 and WRF-AAM, to extend the scope for studying aerosol evolution. Our analysis primarily focuses on examining the variations in observed SSA and AE in relation to the modelled age of the aerosols. This work is unique and builds upon the previous research over the SEA by including AERONET observations over land which allows for a more spatially extensive study of BBA evolution than those prior.

The WRF-AAM model, utilizing CO tracers, provided estimates of the aerosol age (defined as time elapsed since emission). We were able to assess how BBA optical properties evolved during long-range transport over the SEA by tracking the CO tracers in the model for a period of 2 weeks. Our result showed a longitudinal variation in SSA, that corresponds with the model-derived aerosol age and aligns with the transport pathway from the continent. The results revealed that SSA generally increased as BBA were transported away from the emission source, with lower SSA values measured at the Mongu and Huambo sites than those measured at the coastal Namibe site. Throughout 2016, 2017, and 2018, there was a clear trend in mean SSA values, generally increasing with distance from the emission source while over land, with low values recorded at Mongu and Huambo, and the highest observed near the central African coast.

We next applied a model-based vertical separation and AE-based threshold to focus our analysis on the upper-level BBA observations over the SEA. Our analysis showed a distinct temporal trend in FT SSA, where mean FT SSA values initially increased from 0.84 to 0.87 during the first 7 days following emission, followed by a decline back to 0.84 after about 12 days. This implies that the observed aerosol aging led to a 0.03 change in SSA, a trend consistent with in situ measurements reported during the ORACLES campaign by Sedlacek et al. (2022) and Dobracki et al. (2023). In Dobracki et al. (2023), a similar SSA change of approximately 0.06 was observed following a concomitant reduction in OA:BC mass ratio. Given that a variation of ±0.03 in SSA can result in a difference of up to ~20 Wm-2 in local direct radiative effect (Wilcox, 2012), the observed SSA changes in our analysis can have substantial effects on Earth's radiative budget, particularly in the SEA, especially considering the influence of BBA above clouds on the sign and magnitude of TOA forcing over dark ocean surface, as noted by Keil and Haywood (2003). To our knowledge, this study is the first to use remote sensing observations to document changes in BBA properties during long-range transport and associated aging.

We conclude from our analysis that BBA undergo variations in their optical properties that have important implications for the radiation balance. Upon emission, BBA are characterized by low SSA values (high absorptivity), and elevated AE values, indicating the prevalence of small sized particles. During initial stages of transport, BBA SSA increases, signifying a reduction absorptivity, accompanied by a decrease in AE, indicative of particle growth. However, there is a subsequent decrease in SSA (increase in absorptivity) as BBA continue to age during extended transport. This evolution is driven by atmospheric processes. Dobracki et al. (2023) attributed these changes in SSA to various chemical and physical transformation of BBA in the atmosphere.





These results, in agreement with the findings of Dobracki et al. (2023) and Sedlacek et al. (2022), emphasize the continuous evolution of BBA optical properties, influenced by changes in their chemical and microphysical properties. Accurately capturing these evolving properties throughout their lifecycle represents the next step for improving model fidelity and predictive capability. To the best of our knowledge, this study is among the first efforts to investigate BBA evolution over an extended temporal scale, spanning weeks, using columnar observations from remote sensing.

The data analysis techniques and findings from this study contribute to a greater understanding of how BBA optical properties change and the radiative effects associated with those changes. This research also provides further insight into the spatial and temporal evolution of BBA. More importantly, the changes in BBA optical properties associated with aging documented here and elsewhere will need to be incorporated into ESMs to properly represent BBA aerosol properties and effects, and to properly predict future changes in BBA climatic impacts.



**Data Availability:**

The NASA P-3 aircraft data was published by ORACLES Science Team (2021) and can be accessed at these links:

ORACLES 2016: https://doi.org/10.5067/Suborbital/ORACLES/P3/2016_V3;

ORACLES 2017: https://doi.org/10.5067/Suborbital/ORACLES/P3/2017_V3;

ORACLES 2018: https://doi.org/10.5067/Suborbital/ORACLES/P3/2018_V3;

AERONET inversion products are available at https://aeronet.gsfc.nasa.gov/

**Author Contributions**

This study was conceptualized by JR. JR, CJF and AAF formulated the methodology. PES and CH ran the WRF simulations and provided model output. KP, SEL, MSR operated the 4STAR instrument aboard the P-3 aircraft and

processed the 4STAR data. LM contributed to the processing of 4STAR retrievals. AAF organized all datasets, performed analyses, and visualized the results. PG and EL are the PIs for the AERONET network. JR and PZ led efforts to acquire funding and were the PIs for the ORACLES mission. AAF wrote the draft with contributions from all authors.

**Competing Interests**

At least one of the (co-)authors is a member of the editorial board of Atmospheric Chemistry and Physics.

**Acknowledgements**

Portions of the computational work for this paper were supported by and conducted using resources at the University of Oklahoma (OU) Supercomputing Center for Education & Research (OSCER). We extend our gratitude to the entire

NASA ORACLES team for their successful deployment. Our thanks also go to the site PIs and managers for the AERONET stations at Mongu, Huambo, Namibe and Ascension Island. Data analysis and visualization were carried out using open-source libraries in MATLAB and Python.

**Financial Support**

This research has been supported by the University of Oklahoma (OU) start-up package (grant no. 122007900). The

ORACLES field campaign was funded through the NASA Earth Venture Suborbital-2 program (grant no. NNH13ZDA001N-EVS2). PZ acknowledges funding support from DOE ASR award DE-SC0021250.



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
