# Peer review of "Atmospheric processing and aerosol aging responsible for observed increase in absorptivity of long-range transported smoke over the southeast Atlantic"

_EGUsphere, 2024_

## Referee Comment (RC1)

The manuscript titled "Atmospheric processing and aerosol aging responsible for observed increase in absorptivity of long-range transported smoke over the southeast Atlantic" provides an in-depth analysis of impacts of aerosol aging on the single scattering albedo (SSA) and Ångstrom exponent (AE) by using the observation network composed of AERONET (AErosol RObotic NETwork), 4STAR (Spectrometers for Sky-Scanning, Sun-Tracking Atmospheric Research), etc. through the long-range transported smoke over the southeast Atlantic.

The manuscript deals with important scientific themes, however, there are several points that should be addressed to improve its rigor and clarity:

1. In the lines 107 - 110, why single scattering albedo (SSA) increasing indicating their ability to absorb sunlight?

2. In the lines 144 and 530, unclear "OA" and "ASI". define them.

3. In the lines 159 - 169, the introduction of the structure of the paper is not needed.

4. In the Figure 2, We are more concerned about the flights used in the paper. Could they be marked out?

5. In Section 2 "Data and Methods", the introduction is overly lengthy. For example, lines 322 - 332 can be briefly introduced.

6. In the lines 398 - 399, "In Figure 3(b), the vertical distribution of aerosol extinction revealed maximum extinction below 1 km", 1km or 10km? The description doesn't match the figure. Does "aerosol extinction" refer to that of a certain layer or cumulative extinction?

7. In the lines 437 - 438, "We do this by integrating the available collocated AERONET, 4STAR and WRF-AAM output datasets from September 2016, August 2017, and October 2018." Does the data integrated by this method exclude the impact of annual variations on the data?

8. In the Figure 5, Whether the SSA of the two data points of 4STAR that are lower than 0.75 in 8 - 10 days are abnormal data or not, and whether the final results will be affected if they are removed.

9.  In the Figure 7, Is it because the standard deviation (std) is too small that there are no error bars for the data points in the 2 - 4 days and 4 - 6 days, or is it that there is only one data point in each of these two time periods, making it impossible to calculate the error bars?

10.  In the Figure 8 and Figure 9, It is recommended to keep the order of the labels on the x-axis consistent with that of Figure 7 to improve readability. Moreover, Figure 9 and Figure 8 can be combined into one figure.

11.  In Section 3.3.3 Free Tropospheric Single Scattering Albedo, explain why the SSA in the free troposphere (FT) is lower than that in the total column (TC).

12.  In the lines 551 and 553 - 554, "continually decreases from 0.87", "begins at 0.84 (0-2 days), increases to 0.857 (2-4 days), 0.862 (4-6 days), and peaks at 0.871 (6-8 days), before declining to 0.84 (8-10 days) and 0.845 (10-12 days)", increase from 8 - 10 days to 10 - 12 days is in contradiction with "continually decreases".

---

## Author Response (AR1)

**Response to Open Discussion – Community Comment CC1 comments on the manuscript:**

**egusphere-2024-3197**
**Atmospheric processing and aerosol aging responsible for observed increase in absorptivity of long-range transported smoke over the southeast Atlantic.**

CC: Community Comment          RC: Reviewer Comment          AR: Author Response

**CC1: Comments by Dorothy Lsoto -**  **https://doi.org/10.5194/egusphere-2024-3197-CC1**

**CC1: This is a very excellently written abstract and must congratulate the team on a very important research project. It would help to replace "Near the source" with the actual source you are referring to if space allows. This might help keep the reader reminded of what they are still reading about especially for those not very familiar with the science. Overall, this study is very well done from just reading about the novel methodology used.**

AR: We thank Dorothy Lsoto for their gracious comments on our work. Given ACP's word count requirement for the abstract, we unfortunately cannot add any more description to the sentence. We, however, draw the reader's attention to the preceding sentences that read

> "This study applies a novel method combining remote sensing observations with regional model outputs to investigate the aging of the BBA and its impact on the optical properties during transatlantic transport **from emission sources in Africa** to the SEA. Results show distinct variations in Ångstrom exponent (AE) and single scattering albedo (SSA) as aerosol age. Near the source, fresh aerosols are characterized by low mean SSA (0.84) and high AE (1.85), indicating smaller, highly absorbing particles……."

as they highlight the sources to be within the continent of Africa.

**Response to Open Discussion – Community Comment CC2 comments on the manuscript:**

**egusphere-2024-3197**
**Atmospheric processing and aerosol aging responsible for observed increase in absorptivity of long-range transported smoke over the southeast Atlantic.**

CC: Community Comment                    RC: Reviewer Comment                    AR: Author Response

**CC2: Comments by Anoruo Chukwuma - https://doi.org/10.5194/egusphere-2024-3197-CC2**

AR: We would like to thank Anoruo Chukwuma for taking the time to read through our work and provide their feedback. Our response to specific comments is as follows.

**CC2: The present form of the draft is not clear enough and several paragraphs are not complete. Some terms were used in a strange way, for example "450 science...".**

AR: The comment lacks clarity on what aspects of our submission are deemed "not clear enough" or which paragraphs are considered "not complete." Regarding the cited example '450 science…", the full sentence reads

"A total of 56 research flights:12 from the ER-2, only in 2016, and 44 from the P-3B across the three deployments, shown in Figure 2, with over 450 science flight hours collected data on aerosol optical properties (Pistone et al., 2019; Redemann et al., 2021)."

So, it is unclear what is considered 'strange' about this sentence.

**CC2: The method section not clear and several instruments (data) used not mentioned. In general, the draft should be logically linked to tell nice story about method and how AOD from different instruments were subset. For example, subsection 2.6 line 295.**

AR: All instruments, data, observations, and model outputs used in the study are described, with additional information provided in the supplemental material where applicable. AERONET provides AOD retrievals at four wavelengths as discussed in subsection 2.2 while AOD retrieval using 4STAR is described in subsection 2.3. For further technical details on the retrieval methods, we refer you to the cited literature in both sections, as such specifics fall outside the scope of this study. As your concern regarding the cited example is unclear, we have included the relevant excerpt from subsection 2.6 (lines 293–299) below.

"The AERONET inversion products provide AOD and SSA at four wavelengths: 440, 675, 870, and 1020 nm. The relationship between AOD and wavelength, defined by the Ångström exponent formula (Eck et al., 1999) is used to compute EAE($\alpha$):

$$EAE(\alpha) = - \frac{\ln (\tau_1) - \ln (\tau_2)}{\ln (\lambda_1) - \ln (\lambda_2)}, \tag{2}$$

Where $\tau_1$ and $\tau_2$ are AOD at wavelengths $\lambda_1$ and $\lambda_2$. In AERONET and 4STAR retrievals, EAE($\alpha$) is calculated from AOD measurements at 440 nm and 870 nm."

**CC2: I have doubt about use of level 1.5 (not screened) complemented with level 2.0.**

AR: AERONET level 1.5 data are cloud-screened and quality-controlled while level 2.0 data are cloud-screened, quality-controlled and quality-assured: https://aeronet.gsfc.nasa.gov/. Our method includes level 1.5 data only when level 2.0 is not available to maintain a sufficient sample size. The figure below compares AOD at 440 nm from level 1.5 and level 2.0 when they are both available.

[Figure]

[Figure]

**Response to Open Discussion – Anonymous Reviewer RC1 comments on the manuscript:**

**egusphere-2024-3197**
**Atmospheric processing and aerosol aging responsible for observed increase in absorptivity of long-range transported smoke over the southeast Atlantic.**

CC: Community Comment                    RC: Reviewer Comment                    AR: Author Response

**RC1: Comments by anonymous reviewer -  https://doi.org/10.5194/egusphere-2024-3197-RC1**

AR: We sincerely thank the reviewer for the time and effort to provide their feedback on this manuscript. We especially appreciate the helpful comments and positive reception to our work.

**RC1: 1. In the lines 107 - 110, why single scattering albedo (SSA) increasing indicating their ability to absorb sunlight?**

AR: We thank the reviewer for their comment. The sentence "*BBA in this region typically have SSA values ranging between 0.7 and 0.9 from observations (Dubovik et al., 2002; Eck et al., 2013; Haywood et al., 2003), increasing from July to November (Eck et al., 2013), with an average value of 0.85 during the burning season (Dobracki et al., 2023; Eck et al., 2013; Leahy et al., 2007; Pistone et al., 2019), indicating their ability to absorb sunlight.*" was attempting to communicate two points - first is that the seasonality of the SSA in the region shows increasing values from July through November, with July having the lowest SSA value during the burning season, and second that the average value of 0.85 during the burning season is indicative of their significant ability to absorb sunlight. We have revised the paragraph as follows to ensure more clarity.

> "BBA in this region typically have SSA values ranging between 0.7 and 0.9 from observations (Dubovik et al., 2002; Eck et al., 2013; Haywood et al., 2003). These values generally increase from July to November (Eck et al., 2013), with an average value of 0.85 during the burning season (Dobracki et al., 2023; Eck et al., 2013; Leahy et al., 2007; Pistone et al., 2019), indicating their significant ability to absorb sunlight."

**RC1: 2. In the lines 144 and 530, unclear "OA" and "ASI". define them.**

AR: We erroneously missed defining the abbreviations "OA" and "ASI", which stand for Organic Aerosols and Ascension Island respectively. These definitions have now been incorporated: 'OA' is first defined in line 83 and 'ASI' is replaced with Ascension Island.

**RC1: 3. In the lines 159 - 169, the introduction of the structure of the paper is not needed.**

AR: We have now removed the entire paragraph as recommended. However, considering the significance of the paragraph in helping the readers to effectively navigate the manuscript, we are prepared to reinstate it should the editor allow it.

**RC1: 4. In the Figure 2, We are more concerned about the flights used in the paper. Could they be marked out?**

AR: We have revised the figure to include only flight tracks for the observation used in the study. Figure 2 caption now read

> "Map of the SEA showing the NASA P-3B flight tracks during ORACLES 2016 (cyan), 2017 (white) and 2018 (red) for observation days analyzed in this study. The ORACLES aircraft deployed out of Walvis Bay, Namibia in 2016 and São Tomé and Príncipe in 2017 and 2018. Regional AERONET stations are identified using yellow icons. Background imagery from Google Earth (https://earth.google.com) - Data SIO, NOAA, U.S. Navy, NGA, GEBCO, Image Landsat / Copernicus."

**RC1: 5. In Section 2 "Data and Methods", the introduction is overly lengthy. For example, lines 322 - 332 can be briefly introduced.**

AR: The introductory paragraph for section 2.7 has been shortened and summarized as follows

> "In the SEA, as ocean surface temperatures rise, the BL deepens and decouples, with the BL height (BLH) increasing away from the African coast, from approximately 1300m to 1700m, before transitioning to a cumulus-dominated cloud regime, as explained by Zhang and Zuidema (2019, 2021) and Ryoo et al. (2021). Over land, BBA dominate the deep BL, extending beyond 6 km and are then advected over the SEA above the cloud layer by the FT winds (Ryoo et al., 2021), particularly from August to October during strong South African Easterly Jet (AEJ-S) episodes (Adebiyi & Zuidema, 2016). The radiative effects of aerosols in the SEA region are heavily influenced by the interplay between the aerosol layer and cloud layer (Chang & Christopher, 2017; Zhang & Zuidema, 2021)."

**RC1: 6. In the lines 398 - 399, "In Figure 3(b), the vertical distribution of aerosol extinction revealed maximum extinction below 1 km", 1km or 10km? The description doesn't match the figure.**

AR: Figure 3(b) is shown below. The height is on the y-axis, the time is on the x-axis and extinction is the shading shown in the color bar. From the figure, we see that the maximum extinction is within the lowest 2km, implying that there is more aerosol loading closer to the surface (sea level).

The paragraph has been reworded as follows

> "In Figure 3(b), the vertical distribution of aerosol extinction generally revealed maximum extinction within the lowest 2km. The model demonstrated a consistent inverse relationship between extinction and height, with extinction decreasing as height increased, reaching a minimum above 6 km."

**Does "aerosol extinction" refer to that of a certain layer or cumulative extinction?**

The aerosol extinction in this curtain plot refers to the vertically-resolved extinction and not the cumulative.

[Figure]

**Same as Figure 3 (b): WRF-AAM curtain plot with altitude on the Y-axis and time in UTC on the X-axis, showing the smoke plume age forecast (a) and aerosol extinction (b) along the P-3 track (solid green line) during research flight 10 of the third ORACLES deployment on October 17, 2018.**

**RC1: 7. In the lines 437 - 438, "We do this by integrating the available collocated AERONET, 4STAR and WRF-AAM output datasets from September 2016, August 2017, and October 2018." Does the data integrated by this method exclude the impact of annual variations on the data?**

AR: We acknowledge that the term 'integrating' may be misleading in the context of our methodology. Our intent was to convey that we combined the individual observations from each of the three months of the ORACLES campaign with the corresponding collocated model outputs. We have now replaced 'integrating' with 'combining' in the manuscript. Combining the data does not necessarily exclude the annual variations, however by focusing on the aging of the particle – time since emission from the source – using CO tracer in WR-AAM as discussed in section 2.5, we mitigate the impact of seasonal and annual variability of the data. Moreover, prior studies (Eck et al., 2003) have shown limited variability within the continental source region.

**RC1: 8. In the Figure 5, Whether the SSA of the two data points of 4STAR that are lower than 0.75 in 8 - 10 days are abnormal data or not, and whether the final results will be affected if they are removed.**

AR: We thank the reviewer for their suggestion. We did not consider those data points as abnormal given that they pass our filtering methods and have angstrom exponent values greater than 1.2 (as seen again in Figure 12). Nonetheless, we evaluated the impact of excluding them on the result. The two points constitute only 4% of the total data points (49) within the 8-10 day bin and removing those two points will result in a marginal increase in the mean SSA of approximately 0.005, which does not alter the conclusion regarding the overall decreasing trend in SSA for older particles.

**RC1: 9. In the Figure 7, Is it because the standard deviation (std) is too small that there are no error bars for the data points in the 2 - 4 days and 4 - 6 days, or is it that there is only one data point in each of these two time periods, making it impossible to calculate the error bars?**

AR: The error bars represent the standard error (SE) of the mean which is the standard deviation of the sample distribution within each age bin divided by the square root of the number of data points. The two age bins (2-4 days and 4-6) days have the largest number of data points (425 and 319, respectively) and the smallest standard deviations (std) of 0.02 and 0.03 respectively such that the SE is small with values of 0.001 for the two age bins. The summary statistics of SSA across all the age bins is presented in Table 5.

**RC1: 10. In the Figure 8 and Figure 9, It is recommended to keep the order of the labels on the x-axis consistent with that of Figure 7 to improve readability. Moreover, Figure 9 and Figure 8 can be combined into one figure.**

AR: Again, we thank the reviewer for their recommendation. In response, we have revised Figures 8 and 9 to maintain consistency in the x-axis as suggested. However, we prefer to present Figures 8 and 9 separately to enhance clarity for readers – specifically, we believe that this will help to distinguish between the fraction of FT AOD (Figure 8) and the actual AOD (Figure 9), given the difference in the y-axis.

**RC1: 11. In Section 3.3.3 Free Tropospheric Single Scattering Albedo, explain why the SSA in the free troposphere (FT) is lower than that in the total column (TC).**

AR: The following sentences have been added to section 3.3.3.

> "These values of FT SSA are lower than TC SSA due to the exclusion of the BL portion of the TC, which may include other non-BBA aerosols with higher SSA. This difference between FT SSA and TC SSA over the SEA is further discussed in section 3.4."

**RC1: 12. In the lines 551 and 553 - 554, "continually decreases from 0.87", "begins at 0.84 (0-2 days), increases to 0.857 (2-4 days), 0.862 (4-6 days), and peaks at 0.871 (6-8 days), before declining to 0.84 (8-10 days) and 0.845 (10-12 days)", increase from 8 - 10 days to 10 - 12 days is in contradiction with "continually decreases".**

AR: We have removed the word "continually". Now the section reads

> "After 8 days, FT SSA decreases from 0.87, reaching 0.84 after approximately 12 days of transport. This trend in the mean FT SSA across age bins, summarized in Table 5, begins at 0.84 (0-2 days), increases to 0.857 (2-4 days), 0.862 (4-6 days), and peaks at 0.871 (6-8 days), before declining to 0.84 (8-10 days) and 0.845 (10-12 days)."

**Response to Open Discussion – Anonymous Reviewer RC2 comments on the manuscript:**

**egusphere-2024-3197**
**Atmospheric processing and aerosol aging responsible for observed increase in absorptivity of long-range transported smoke over the southeast Atlantic.**

CC: Community Comment                RC: Reviewer Comment                AR: Author Response

**RC2: Comments by anonymous reviewer -** https://doi.org/10.5194/egusphere-2024-3197-RC2

**General Comments:**

Fakoya et al. show a method to predict the evolution of optical properties and vertical aerosol composition upon atmospheric aging in the southeast Atlantic (SEA) region. They used remote sensing data combined with a regional model. They found that biomass-burning aerosol absorption first decreases and then increases with aging. Overall, this well-designed study can help reduce the current uncertainties in aerosol climate effects. I have a few minor comments to help improve this paper. Please see my comments below:

AR: We appreciate the reviewer for their time and positive review of our paper. We believe that their constructive feedback has helped improve the overall readability of our paper. First, we would like to note that we have accepted their suggestion (comment #2) on using EAE instead of AE as the abbreviation for Extinction Ångstrom Exponent, and we have updated our responses accordingly.

**RC2: 1. It is unclear how you developed the AE threshold. Did you do any statistical tests to verify your threshold? Why do you choose these four values to test?**

AR: We developed the threshold based on the knowledge of the EAE distribution within the source regions from literature (Eck et al., 1999; Eck et al., 2013). However, we tested the different EAE values to account for potential changes in the EAE of aging biomass-burning aerosol due to particle growth by condensation or coagulation (Eck et al., 2023) as discussed in section S3 of the supplemental material: (https://egusphere.copernicus.org/preprints/2025/egusphere-2024-3197/egusphere-2024-3197-supplement.pdf). We did not conduct statistical tests, but we decided on the optimal threshold AE value of 1.2 as a balance between the BBA EAE and the sample size over the ocean. When we applied EAE filters of lower value (0.75 and 1.0), we observe elevated values of SSA for aged aerosols (8-12 days). When we applied an EAE filter of higher value (1.4), we see lower values of SSA for aged aerosols (8-12 days) but fewer observations in those age bins.

**RC2: 2. I suggest using EAE for Extinction Angstrom Exponent across the whole manuscript instead of AE since AE might be confused with AAE (Absorption Angstrom Exponent).**

AR: We have complied with this suggestion and updated the entire manuscript and the supplemental material to define Extinction Ångstrom Exponent as EAE.

**RC2: 3. How did the AE vary along with the aerosol absorption properties?**

AR: The relationship between EAE, aerosol age and SSA in the total column and free troposphere is shown in figures S14 and S15 of the supplement respectively, with the mean values visualized in the figures provided below. From both figures, we can see that EAE decreases with an increase in SSA and with age. With respect to the SSA, in the TC and

the FT, high values of EAE correspond to low SSA values within the first 2 days of aging, with a continuous decrease in EAE corresponding to an increase in SSA between 2-8 days. After 10 days, there is a slight increase in mean EAE in the TC but not in the FT, which may be attributed to the presence of non-absorbing fine-mode marine aerosols in the boundary layer, which is captured in the TC measurements. This relationship between EAE and SSA also supports our discussion in section 3.4 on the importance of employing the two-pronged method to isolate non-BBA contributions to the evolution.

[Figure]

**Relationship between EAE and SSA with respect to age in the total column (upper panel) and free troposphere (lower panel)**

**RC2: 4. Fig. 4 is unnecessary since the same information is shown in Fig. 6.**

AR: We concur and have removed Figure 4. Figure 6 is now designated as Figure 4 and all text references have been updated accordingly.

**RC2: 5. Have you considered lensing enhancement since SOA can condense on soot and tar balls?**

AR: Given the nature of remote sensing observations we have used in this study, we have made a previously unstated assumption of only external mixing of the particles over the ocean to facilitate the separation of marine aerosols from BBA. This assumption is now explicitly mentioned in the first sentence of section 2.7.1, which reads

> **2.7.1 Application of model-derived extinction ratio**
>
> To address the potential influence of MBL aerosol properties on the TC observations by AERONET and 4STAR over the SEA, we use a model-derived vertical distribution of extinction to estimate the FT contributions to the TC measurements, while assuming an external mixing state of particles over the ocean."

We contextualize the evolution of SSA with respect to chemical and physical changes described by Dobracki et al. (2023) using in situ measurement, as discussed in section 3.4, with the initial increase in SSA corresponding to an increase in OA:BC ratio. We have also now included lensing enhancement as a potential factor responsible for the decrease in SSA of aged particles. The text has been updated accordingly:

> "The changes in SSA presented in this study are primarily associated with chemical and physical processes in the atmosphere (Dobracki et al., 2023). Fresh aerosols over the continent show a low SSA, likely a result of a high proportion of rBC from flaming fires. As these aerosols age in the atmosphere, they accumulate organic coatings, a process that concurrently occurs with homogeneous nucleation of secondary organic aerosol (SOA) from volatile organic compounds (VOCs). These processes happen rapidly within hours and continue for the first few days (Hodshire et al., 2019; Sedlacek et al., 2022), increasing the contribution of OA to the aging particles, thereby increasing SSA. After 6-8 days, the SSA starts to decrease, possibly due to lensing enhancement or a reduction in the production of SOA. Additionally, heterogeneous oxidation may drive the repartition of aerosol mass back into gas phase. In addition to compositional changes in BBA driven by accumulation and/or evaporation of organic coatings, a concurrent change in fine-mode particle size is also likely during aging. These size changes are possibly a consequence of the same chemical processes and contribute to the observed trends in optical properties. For instance, as particles shrink, their scattering efficiency decreases, which in turn can lead to a reduction in SSA"

**Specific comments:**

**RC2: For Fig. 1, it is not clear what each green line represents. Are they from different original days? Moreover, it might be good to add labels of back days or transport directions on the path. It will be good to add vertical transport as well. Also, I suggest labeling flight directions on the red line. For panel b, I recommend adding the y-axis of altitude. Also, please increase the font size. They are too small.**

AR: The green lines all represent back trajectories ending on October 18, 2018, at different altitudes between 4000m and 5000m to intersect with the P-2 flight track on October 17, 2018. We provide a low-fidelity figure of the trajectories in the supplemental material – Figure S1b (copied below) that shows the altitude profile of the trajectories.

[Figure]

[Figure]

**Same as Figure S1(b): 7-day backward trajectories ending over the SEA on October 18, 2018. The trajectories (b) intersect the P3 flight track on October 17, 2018, shown in Figure 1 about 12:00-13:00 UTC between 4000 – 5000m.**

We have now updated Figure 1a and edited the caption to include the description of the trajectories as follows,

> "Satellite image showing the southeast Atlantic Ocean covered by the stratocumulus cloud deck with smoke being advected over it. The smoke from continental fires (orange dots) is being

transported, evident by the 7-day backward HYSPLIT trajectories (green) ending between 4000m – 5000 m on October 18, 2018, to intersect the P-3 flight (red) on October 17, 2018. Yellow icons represent AERONET stations selected for this study. https://worldview.earthdata.nasa.gov/, https://www.ready.noaa.gov/HYSPLIT_traj.php, https://earth.google.com, https://aeronet.gsfc.nasa.gov/).”

The P-3 flight on October 17, 2018, conducted multiple routines along the same paths which does not allow us to provide a specific direction on Figure 1a, However, we provide a supplemental figure (Figure S10) and copied below, that details direction and time of the flight. The HYSPLIT back trajectories intersect the P3 flight track on October 17, 2018, shown in Figure 1 about 12:00-13:00 UTC between 4000 – 5000m, approximately 4-6 days after departure from the source region.

We have updated the panel b figure to include a y-axis label of altitude, and we have increased the font size to improve readability.

[Figure]

**Same as Figure S10: NASA P3-B aircraft flight track over the SEA on October 17, 2018**

The updated Figure 1 is shown below

[Figure]

[Figure]

**RC2: It is unclear which tracer gases you used to estimate the aging time. Did you use CO? Then, it should be singular, not plural. Which emission inventory did you use for your model simulation?**

AR: Thank you for your suggestion. We have addressed the age tracer in L256 and the paragraph now reads

> "To estimate the age of the aerosols, defined as the time since emission, we utilized carbon monoxide (CO) tracer coupled with smoke emissions within the WRF-AAM model."

The emission inventory for both the WRF-AAM and WRF-CAM5 simulations is the Quick-Fire Emission Dataset version 2 (QFED2).

**RC2: Equation 4. It should be RFT, not RBL, on the left side.**

AR: Thank you, we had erroneously written $R_{BL}$ instead of $R_{FT}$. The equation is now updated.

**RC2: Please label wavelengths when you discuss or show anything related to optical properties such as SSA, AOD, and AE.**

AR: We have updated the figures and the text within the manuscript accordingly to include the wavelengths for the optical properties where applicable.

**RC2: L308, please define FT when it first appears in the main text.**

AR: We have defined the FT as free troposphere.

**RC2: L548-551, "These distributions … SSA in the region." I might miss something, but it is unclear to me how you come up with this conclusion. Could you explain a little bit more?**

AR: Based on our analysis to evaluate the contribution of observations from each month of the ORACLES campaign (August, September, October) to the age bins, as detailed in Table S1 and S2 of the supplemental material and copied below, we observe several patterns in the distribution of the retrievals.

**Table S1: Total count of FT SSA by month for [8-10] and [10-12] day age group at (AE ≥ 1.2)**

| Age Group | August | September | October | Total |
|---|---|---|---|---|
| 8-10 | 21 | 25 | 3 | **49** |
| 10-12 | 6 | 17 | 0 | **23** |
| **Total** | **27** | **42** | **3** | **72** |

**Table S2: Total count of FT SSA by station for [8-10] and [10-12] day age group at (AE ≥ 1.2)**

| Age Group | Ascension | Namibe | Huambo | Mongu | 4STAR | Total |
|---|---|---|---|---|---|---|
| 8-10 | 6 | 6 | 0 | 0 | 37 | **49** |
| 10-12 | 11 | 0 | 0 | 0 | 12 | **23** |
| **Total** | **17** | **6** | **0** | **0** | **49** | **72** |

For October (2018), there are no observations in the 10-12 day bin and only 3 data points in the 8-10 day bin, accounting for less than 2% of total observations (229) for that year. In the 8-10 day bin, retrievals are nearly evenly distributed between August (2017) and September (2016), while the 10-12 day bin contains approximately three times as many retrievals from September compared to August. These contributions, (27) from August (2017) and (42) from September (2016), represent about 5% and 16% of their respective total observations (556 and 258). This distribution of observations in the older (8-10) and (10-12) day bins suggests that the decrease in SSA observed for older aerosol plumes is not due to an overrepresentation of August data, which climatological studies suggest exhibit lower SSA due to their emissions earlier in the burning season. Instead, the observed trend is more consistent with chemical aging processes, agreeing with prior in situ studies.

We edited the paragraph below to enhance clarity.

> "The analysis showed that for October (2018), there are no observations in the 10-12 day bin and only (3) data points in the 8-10 day bin. In the 8–10 day bin, retrievals are nearly evenly distributed between August (2017) and September (2016), while in the 10–12 day bin, the ratio of September to August retrievals is approximately 3:1. The total number (3) of observations from October (2018) in the 8-10 day bin (3) and 10-12 day bin (0) represent less than 2% of total observations (229) for that year. Meanwhile, the contribution (27) from August (2017) and (42) from September (2016) represent about 5% and 16% of the total observations (556 and 258, respectively) for the years. This distribution of observations in the 8-10 and 10-12 day bins suggests that the decrease in SSA observed for older aerosol plumes is not due to an overrepresentation of August data, which climatological studies suggest exhibit lower SSA due to their emissions earlier in the burning season. However, other factors, rather than the timing of the emission, drive the observed SSA change in aged plumes."

**RC2: L564-566, "This decrease in … of larger particles." This is also unclear to me.**

AR: Our aim here is to communicate the importance of combining both the model-based vertical partitioning (Section 2.7.1, L326-355) with the EAE threshold filtering (section 2.7.2, L356-368) to allow for a more precise discrimination of biomass-burning aerosols, ensuring that non- or less-absorbing and larger aerosols are excluded. We first applied the EAE filtering to the total column (TC) observation over the ocean (Figure 7[6]) to assess the variation in mean SSA within the TC by just the removal of larger particles (EAE < 1.2). We then applied the EAE filtering to the free troposphere (FT) data after partitioning. This was because, after partitioning, there were a few samples with EAE << 1.2, as shown in the figure below (not provided in the manuscript).

[Figure]

**Same as Figure 5 but for the FT: The relationship between SSA532nm (y-axis), EAE (color bar), and aerosol age (x-axis) in the free troposphere (FT). The different markers represent the site of observation while the marker shading represents the EAE.**

We then compared the mean TC SSA to the mean FT SSA after EAE filtering (Figure 13[12]), which reveals a pronounced decrease in mean FT SSA, up to 2%, relative to the mean TC SSA for older aerosols. This change suggests that by combining the two methods described in section 2.7, we are able to achieve our desired isolation of fine, BBA particles in the FT from the contribution of larger (EAE < 1.2), non-absorbing aerosols. Applying the EAE filtering solely to the TC data removed larger particles but did not account for other less absorbing fine particles (from mixing of BBA with marine aerosols with the boundary layer) or non-absorbing fine-mode marine aerosols. However, the application of the EAE filtering to the FT data after partitioning efficiently isolate all non-BBA particles.

The paragraph now reads:

> "We examined the mean FT SSA against the mean TC SSA at the established optimal EAE threshold (Fig. 12) to gain a clearer understanding of the evolution of SSA shown in Fig. 12 and to highlight the significance of employing the combination of the model-based vertical extinction separation with an EAE filter. The results demonstrate that FT SSA decreases by more than 2% for BBA aged beyond 8 days. This decrease in mean FT SSA compared to TC SSA suggests that by combining the two methods described in section 2.7, we are able to achieve our desired isolation of fine, BBA particles in the FT from the contribution of larger (EAE < 1.2), non-absorbing aerosols. Applying the EAE filtering solely to the TC data removed larger particles but did not account for other less

absorbing fine particles, from mixing of BBA with marine aerosols with the boundary layer (Dang et al., 2022), or non-absorbing fine-mode marine aerosols (Fitzgerald, 1991). However, the application of the EAE filtering to the FT data after partitioning efficiently isolate all non-BBA particles. Overall, the combined use of both techniques, reveals a distinct evolution pattern of BBA in the FT. To examine the sensitivity of our assumption that $SSA_{BL} = 1$, we tested alternative $SSA_{BL}$ values to account for varying degrees of absorption within the BL. The results showed an evolution of FT SSA similar to that in Figure 12, with a decrease in mean FT SSA after 8 days (supplemental Figure S6)."

**RC2: L578, "These processes happen rapidly … first few days," Needs citations.**

AR: We have added appropriate citation and references to the sentence. It now reads

"These processes happen rapidly within hours and continue for the first few days (Hodshire et al., 2019; Sedlacek et al., 2022), increasing the contribution of OA to the aging particles, thereby increasing SSA."

**RC2: L579, "heterogeneous oxidation repartitions some of the aerosols …" Oxidation refers to a chemical transformation, whereas repartition suggests a physical or phase-based process. It would be helpful to clarify whether the oxidation itself leads to repartitioning through chemical breakdown, or if the repartitioning occurs as a result of phase equilibrium shifts independent of oxidation.**

AR: In the free troposphere, evidence suggests that organic aerosol (OA) mass concentration decreases as aerosols age, as indicated by increasing $f44$ values and aerosol age estimates from WRF for plumes with similar rBC: $\Delta$CO ratios (Dobracki et al., 2023). Additionally, Dobracki et al. (2023) observed minimal variation in OA mass concentrations with changes in relative humidity (RH) (figure 15), while more recent findings (Dobracki et al., 2025) suggest a potential dependence on ambient temperature during decent into the marine boundary layer (MBL) (their supplementary figure 6). However, in the free troposphere, the weaker temperature gradient makes phase equilibrium shifts unlikely to drive additional repartitioning of OA. This supports the explanation that the decrease in OA mass concentration for aged plumes is primarily due to oxidation processes rather than thermodynamic effects.

**RC2: L581, it is unclear what is the concurrent change you mentioned here.**

AR: What we meant by the concurrent change in the fine-mode particle size is that as the simultaneous changes in composition and optical properties of the aerosols occur, driven by chemical processes, there is a possible corresponding change in particle size that contributes to the SSA changes. For example, fine mode scattering efficiency will decrease with a decrease in particle size and can also cause slight lowering of the SSA. We revised the sentence to improve the clarity.

[revised manuscript text omitted]

---

## Author Response (AR2)

**Response to Editorial Comment on the manuscript:**

**egusphere-2024-3197**
**Atmospheric processing and aerosol aging responsible for observed increase in absorptivity of long-range transported smoke over the southeast Atlantic.**

EC: Editor Comment          RC: Reviewer Comment                    AR: Author Response

**EC: Public justification (visible to the public if the article is accepted and published)**:
I want to take this chance to say thanks to the reviewers for the valuable comments. I also appreciate the authors being serious about the comments. After reading all of the materials, I have some additional suggestions for this manuscript:

**EC: 1. This manuscript is generally lengthy, especially the method part already reaches line 372. Though the text helps to demonstrate the broad background knowledge of the authors, it does not help to keep readers focused on the important scientific findings. Here are several specific suggestions for your consideration:**

> **1) Delete the text regarding the introduction of the structure of the paper,**

> AR: We have complied with your recommendation and have removed the paragraph accordingly.

> **2) The first few sentences introducing the measurement modes of the 4STAR. You are not using the other two modes at all. Start directly from "This study…",**

> AR: We appreciate your suggestion and have revised the section. Section 2.3 now reads:

**2.3 Airborne measurements**

This study uses sky radiance measurements from the airborne 4STAR instrument (Dunagan et al., 2013) aboard NASA P-3B aircraft, enabling AERONET-like observations in remote regions beyond ground-based network coverage. Its frequent co-deployment with in-situ instrumentation offers a more comprehensive characterization of aerosol properties. This analysis focuses on observations made in the sky-scanning mode (Pistone et al., 2019), using both ALM and principal plane (PPL) scans. Given that most ORACLES flights occurred near solar noon, limiting the angular range of ALM scans, PPL scans were selectively included if they met specific quality control (QC) criteria. Here, we processed 4STAR sky scans using the QC criteria from Mitchell et al. (2023). These criteria were adapted from Pistone et al. (2019) for four-wavelength 4STAR retrieval of ORACLES 2016-2018 and serve as a proxy for AERONET level 1.5 aerosol inversion standards. The criteria are: (1) AOD (400 nm) > 0.2, (2) altitude difference < 50 m, (3) sky error < 10%, (4) minimum scattering angle < 6°, (5) maximum scattering angle > 50°, (6) mean scattering angle difference < 3° (between 3.5 and 30°), (7) maximum scattering angle difference < 10° (between 3.5 and 30°), (8) roll standard deviation < 3°, (9) passes retrieval boundary test - ensuring that the retrieval is within limits of parameter space, and (10) maximum altitude < 3000 m. A

summary of the valid QC'd 4STAR retrievals of SSA, AOD, EAE (Mitchell et al., 2023) in the ORACLES dataset (Oracles, 2020) for all three deployments is given in Table 1.

**3) The Second paragraph introducing the WRF-CAM5 is a good summary of the model, but I don't think it is necessary here unless you have done something unique to the model. As a method part, the first and last paragraphs are enough.**

AR: We appreciate your suggestion and the section now reads:

**2.4 WRF-CAM5: Concept and Configuration**

WRF-CAM5 is an adaptation of the WRF-Chemistry (WRF-Chem) model (Grell et al., 2005), which integrates the physics and aerosol packages of the global CAM5 (Ma et al., 2014; Zhang et al., 2015a), making it suitable for studying multi-scale atmospheric processes and evaluating aerosol and physics parameterizations in global climate models (Wang et al., 2018). WRF-CAM5 has been widely applied to investigate air quality and climate interactions in Asia and the United States (Campbell et al., 2017; Wang et al., 2018; Zhang et al., 2015b) and has shown good skill in capturing smoke concentration, aerosol properties, and the vertical distribution of BBA in the SEA region (Doherty et al., 2022; Chang et al., 2023). In this study, the model is configured at 36km horizontal resolution with 74 vertical layers over the spatial domain 41°S-14°N, 34°W-51°E, initialized every five days using the National Center for Environmental Prediction (NCEP) Final Operational Global Analysis (NCEP FNL) and Copernicus Atmosphere Monitoring Service (CAMS) reanalysis datasets as detailed in Shinozuka et al. (2020a) and Doherty et al. (2022), with daily smoke emissions  from the Quick-Fire Emissions Dataset version 2 (QFED2) (Darmenov and Da Silva, 2015).

**4) There is a long introduction on why to separate the BL contribution from TC. The sentence "Dang et al. (2022) showed that BBA sampled during ORACLES dominate… " is good enough, and all of the text before can be potentially removed.**

AR: We have revised the introduction to section 2.7 on why we separate the BL contribution from TC. In line with your recommendation, we have removed portions of the text while retaining a few sentences to briefly describe the transport of BBA by the AEJ-S from the continent to the maritime atmosphere over the SEA. The revised section now reads:

**2.7 Separating Boundary Layer (BL) contributions from Total Column (TC) observations**

As sea surface temperatures increase, the boundary layer (BL) deepens offshore before transitioning to a cumulus regime (Zhang and Zuidema, 2019, 2021; Ryoo et al., 2021). Continental BBA are lofted above 6 km and advected above the cloud layer by free troposphere (FT) winds (Ryoo et al., 2021), particularly during strong south African Easterly Jet (AEJ-S) episodes (Adebiyi and Zuidema, 2016). Although generally elevated, subsidence (Wilcox, 2010) and low-level easterlies (Diamond et al., 2018) can entrain BBA into the BL between June and August, altering aerosol properties (Dobracki et al., 2025).

Dang et al. (2022) showed that BBA sampled during ORACLES dominate the FT while sea salt aerosols may often dominate the BL over the SEA with a fraction of BBA mixed with sea salt aerosols in the BL. Therefore, our goal of investigating the evolution of BBA from TC observations is complicated by the potential contribution of non-BBA aerosols from the MBL. To address this, and given that AERONET and 4STAR provide columnar retrievals above the observation altitude, we employed a two-pronged approach, detailed in Section 2.7.1 and 2.7.2, to isolate the FT aerosol from the columnar observations. First, we applied a model-derived ratio to partition aerosol loading in the FT and BL over the SEA. Subsequently, we implemented a size thresholding technique to exclude contributions from larger particles, ensuring our analysis remains focused on BBA properties.

**EC: 2. RC2 5 and text in the revised manuscript: There are several problems in this revised paragraph.**

**1) Why emphasize flaming here? Smoldering is more efficient in producing rBC and brown carbon.**

**2) Organic coating is the condensation of organic vapors on the existing particles, while nucleation produces new particles. Don't mix them.**

**3) Many lab experiments have shown that the biomass burning BrC has a lifetime of hours to 2 or 3 days. This bleaching process might be the main reason for the increase in SSA during transport.**

**4) Increasing OA does not mean increasing SSA. It's true only when the OA is not absorbing or less absorbing.**

**5) During 6-8 days, aging probably already causes organic loss in the particles. Therefore, the lensing effect is weaker, which increases SSA. But due to the loss of less absorbing species (compared to black carbon), the mass absorption efficiency increases and size decreases, both lead to decreased SSA. The authors need to carefully change the text.**

AR: We thank you for your insights. We have revised the paragraph as below:

The changes in SSA presented in this study are primarily associated with chemical and physical processes in the atmosphere (Dobracki et al., 2023). Fresh aerosols over the continent exhibit a low SSA, which we attribute to a high proportion of rBC compared to other aerosol components. This interpretation agrees with previous findings further offshore (Denjean et al., 2020b; Wu et al., 2020; Dobracki et al., 2023) and is characteristic of emissions from flaming grassland fires. As these aerosols age in the atmosphere, they accumulate an organic coating, a process that begins rapidly within hours and continues for the first few days (Hodshire et al., 2019; Sedlacek et al., 2022), increasing the contribution of OA to the total mass of the aging particles. Unlike field measurements over the Atlantic Ocean that are relevant for more aged aerosols, our results show a progressive increase in SSA during the first 6 days of aging. After 6-8 days, however,

the SSA starts to decrease, consistent with in-situ observations over the maritime atmosphere. We hypothesize that this decrease in SSA is due to increased absorption per particle from lensing effects, consistent with Taylor et al. (2020). Moreover, heterogeneous oxidation may drive the repartition of aerosol mass back to the gas phase, reducing the overall OA:BC mass ratio. In addition to compositional changes driven by the accumulation and/or evaporation of organic coatings, there is likely a shift in fine-mode particle size, potentially resulting from the same chemical and physical processes that also influence the optical properties.

**3. The abstract must be one single paragraph.**

AR: The abstract is now combined into a single paragraph.

**4. The authors used indentation for starting some new paragraphs, but not to all. Please be constant.**

AR: Thank you for your suggestion. We have revised the manuscript to consistently apply indentations for new paragraphs.